



# Stratospheric gravity waves excited by Hurricane Joaquin in 2015: 3-D characteristics and the correlation with hurricane intensification

Xue Wu[1, 2], Lars Hoffmann[3*], Corwin J. Wright[4], Neil P. Hindley[4], M. Joan Alexander[5], Silvio Kalisch[6], Xin Wang[1, 2*], Bing Chen[7*], Yinan Wang[1, 2], Daren Lyu[1, 2]

[1]Key Laboratory of Middle Atmosphere and Global Environment Observation, Institute of Atmospheric Physics, Chinese Academy of Sciences, Beijing, China

[2]University of Chinese Academy of Sciences, Beijing, China

[3]Jülich Supercomputing Centre, Forschungszentrum Jülich, Jülich, Germany

[4]Centre for Climate Adaptation and Environment Research, University of Bath, Bath, UK

[5]NorthWest Research Associates, CoRA Office, Boulder, USA

[6]Lab of Atmospheric Dynamics, Department of Atmospheric Science, Yonsei University, Seoul, South Korea

[7]Key Laboratory of Atmospheric Environment and Processes in the Boundary Layer over the Low-Latitude Plateau Region, Department of Atmospheric Science, Yunnan University, Kunming, China

*Correspondence to*: Lars Hoffmann (l.hoffmann@fz-juelich.de), Xin Wang (wangx2003@mail.iap.ac.cn), and Bing Chen (chenbing@ynu.edu.cn)

**Abstract.**

Despite progress, accurately forecasting tropical cyclone (TC) intensity, especially rapid intensification, remains a significant challenge. The correlations between the stratospheric gravity waves (GWs) excited by TCs and TC intensity have been recognized. However, partly due to the limitations of conventional analysis methods and observational filters of current satellite instruments, the characteristics of stratospheric GWs that indicate TC intensification remain unclear. This study examined the specific characteristics of GWs and their linkage to hurricane intensification by high-resolution, realistic model simulations and 3-D wave analysis method. First, the stratospheric GWs excited by Hurricane Joaquin in 2015 were simulated using the Advanced Weather Research and Forecasting (WRF) model. Then, the GW characteristics were analyzed using the novel 3-D Stockwell transform method. The GWs excited by Hurricane Joaquin are in the mid-frequency range and propagate outward from the hurricane center counterclockwise while moving upward in a spiral. A high-level time-lagged correlation exists between the intensities of the hurricane and stratospheric GWs during hurricane intensification, making it possible to detect an increase in hurricane intensity by observing an increase in stratospheric GW intensities. Compared to the weakening period, the stratospheric GWs excited during hurricane intensification exhibit relatively higher frequencies, shorter horizontal wavelengths, and longer vertical wavelengths, with this contrast particularly evident near the center of the hurricane. This study provides further knowledge for potentially monitoring hurricane intensification by observing stratospheric GWs using satellite instruments in the infrared and microwave bands when it is difficult to use other measurement techniques.



**1 Introduction**

Tropical cyclones (TCs) are hazardous meteorological events that consistently result in loss of life and property. Despite a substantial reduction in track forecast errors over the past decade, there has been only a modest decrease in the forecast error of intensity (DeMaria et al., 2014). Particularly, predicting abrupt changes in TC intensity, such as rapid intensification (RI), remains challenging and is rarely successful (Cangialosi et al., 2020). In recent decades, with climate change, TCs have become more intense (Emanuel, 2017; Kang and Elsner, 2019), the frequency of TCs undergoing RI has increased (Bhatia et al., 2019; Balaguru et al., 2018), and TCs that experience rapid intensification have a higher probability of developing into stronger and potentially destructive TCs (Lee et al., 2016). While many studies have investigated changes in TC structure and the influence of environmental conditions before TC intensification (e.g., Chen et al., 2011; Kieu et al., 2014; Onderlinde and Nolan, 2014; Kaplan et al., 2015; Rogers et al., 2017), accurate forecasting of TC intensity changes still faces significant challenges (Doyle et al., 2017).

In addition to being catastrophic weather systems, tropical cyclones also serve as sources of excitation for stratospheric gravity waves (GWs) (Hoffmann et al., 2013). Oscillating drafts and thermal forcing due to latent heat release in TCs are two primary mechanisms for generating GWs (Beres et al., 2002). While GWs excited in the upper troposphere rapidly propagate from the source upward to the stratosphere and even higher levels, the wave amplitudes grow and become even more prominent (Liu et al., 2014). The characteristics of stratospheric GWs excited by TCs (TC-GWs hereafter) may reflect changes in TC intensity as the two mechanisms that excite GWs are also associated with changes in TC intensity. First, extremely strong updrafts (also known as convective bursts) typically intensify in the eyewall up to a few hours before TC intensification (Wang and Wang, 2014; Hazelton et al., 2017). Second, this process leads to deep latent heating in the upper troposphere and the formation of a warm core (Ohno and Satoh, 2015); the balanced dynamics of TC structure and flows towards thermal forcing in the eyewall contribute to TC intensification (Fudeyasu and Wang, 2011; Wang and Wang, 2014).

The connections between the stratospheric GWs and the TC intensities are found by idealized model simulations (Nolan and Zhang, 2017) and verified by satellite observations. Hoffmann et al. (2018) utilized approximately 14 years of Atmospheric Infrared Sounder (AIRS) observations of TC-GWs to investigate the correlation between TC-GWs and TC intensity. The study discovered a significant correlation, indicating that TC-GWs are more intense and active during TC intensification. The frequency of TC-GWs during TC intensification was twice as high as during TC weakening. Using 16 years of multiple sources of satellite observations, Wright (2019) observed a similar phenomenon: GW amplitudes increase before the peak intensity of TCs, followed by a sudden decrease afterward. Due to the "observational filter" effect, none of the presently available instruments can comprehensively detect the entire spectrum of GWs (Alexander and Barnet, 2007; Wright et al., 2015). Therefore, it is essential to further investigate the connections between the stratospheric GWs and the TC intensities with realistic numerical simulations. Wu et al. (2022) conducted high-resolution realistic model simulations



focusing on a specific case, the Hurricane Joaquin, which is also the subject of the present study. They verified the
significant correlations between TC-GWs and TC intensity that revealed by theoretical model simulations and satellite
observations, and found that GW activity is more frequent and intense during hurricane intensification, particularly for the
most intense GW activity.
Observing stratospheric GWs offers an advantageous method for inferring changes in TC intensity, particularly when
cloud canopies obscure the TC eye and eyewall from instruments using visible and infrared bands. Motivated by the critical
need to monitor and accurately predict TC intensification, recent research has increasingly focused on using satellite
instruments in the infrared and microwave bands for observing TC-GWs, thereby enabling more precise inferences about TC
intensity evolution (Miller et al., 2018; Tratt et al., 2018). However, the mechanism that cause the correlation between TC-
GWs activity and TC intensification is not yet thoroughly understood, and the specific characteristics, such as wavelengths of
the TC-GWs that indicate TC intensification are not clear.
Building upon Wu et al. (2022), this study extends beyond their emphasis on validating the correlations between the
intensities of stratospheric GWs and the hurricane. It delves deeper into examining the characteristics of GWs, especially their
linkage to the intensification phases of the hurricane. First, we conducted high-resolution numerical simulations of a
challenging case, Hurricane Joaquin (2015). Hurricane Joaquin is a representative case of difficulties in hurricane intensity
forecasts. The official intensity forecast errors for Hurricane Joaquin exceeded the mean official errors during the preceding
five-year period at all forecast intervals (Berg, 2016). Then, our study examines the 3-D characteristics of the simulated TC-
GWs. The characteristics of TC-GWs have been widely analyzed based on numerical simulations and observations (e.g., Kim
and Chun, 2010; Chen et al., 2012; Jewtoukoff et al., 2013; Chane Ming et al., 2014; Yue et al., 2014; Wu et al., 2015).
Nevertheless, the conventional 1-D and 2-D wave analysis method can only reveal the wavelengths and frequency distribution
in a domain but lacks the ability to pinpoint the specific location or timing of distinct characteristics. In this study, a novel 3-
D Stockwell transform method is employed to estimate the characteristics of the stratospheric GWs on 3-D grid points, namely
the dominant wave frequencies, wavelengths, wave speed, and propagation directions. By investigating the features of the
GWs and their connection with the intensification of Hurricane Joaquin, this study will identify the key characteristics and
locations of the stratospheric TC-GWs that indicate hurricane intensification. The goal is to lay the groundwork for monitoring
TC intensification through satellite observations of stratospheric GWs.
The remaining sections are organized as follows: Sections 2 and 3 describe the data and model configuration; Sect. 4
introduces the novel 3-D Stockwell transform method; Sect. 5 presents the results; Sect. 6 discusses the results and gives the
conclusions.



## 2 Data

### 2.1 ERA5 reanalysis

The initial and boundary conditions of the WRF simulation of Hurricane Joaquin were obtained from the ERA5 reanalysis. The ERA5 reanalysis (Hersbach et al., 2020), a fifth-generation product from the European Centre for Medium-Range Weather Forecasts (ECMWF), covers the period from 1940 to the present. The dataset is accessible at 137 vertical hybrid sigma-pressure levels, with the highest level at 0.01 hPa (approximately 80 km altitude) and at the surface level. For our simulations, we utilized hourly data with a horizontal sampling of 0.25°×0.25° to establish initial and boundary conditions.

The ERA5 reanalysis demonstrates an enhanced representation of tropical cyclones (Hodges et al., 2017; Li et al., 2020) compared to the preceding ECMWF reanalysis. It provides improved resolution of convection over both oceanic and continental regions (Hoffmann et al., 2019). This enhancement significantly contributes to the precision of the hurricane intensity simulation.

### 2.2 Tropical cyclone track and intensity archive

This study utilized the International Best Track Archive for Climate Stewardship (IBTrACS) dataset (Knapp et al., 2010) to evaluate the track and intensity of Hurricane Joaquin in the model simulation. The dataset is constructed through subjective satellite-based Dvorak technique intensity assessments from the Tropical Analysis and Forecast Branch (TAFB) and the Satellite Analysis Branch (SAB), in conjunction with the objective Advanced Dvorak Technique (ADT). It provides the TC track and intensity estimates in terms of minimum sea level pressure and maximum sustained winds at intervals of 3 to 6 hours.

## 3 WRF Model configuration

The numerical simulation was performed using Version 3.9.1 of the Advanced Weather Research and Forecasting (WRF) model (Skamarock et al., 2008). The model configuration mainly follows the setup of Wu et al. (2022). The simulation adopted a concurrent one-way nested configuration comprising a stationary outer domain (D01) and an inner nested vortex-following domain (D02). The grid resolutions for D01 and D02 were 12 km and 4 km, respectively, with corresponding time steps of 12 s and 4 s. D01 provided boundary conditions for D02, and no feedback from D02 to D01 was incorporated. The vertical domain covered 90 sigma levels from the surface to 1 hPa (~48 km). Vertical resolution exhibited a gradient, with the finest resolution near the surface gradually decreasing to approximately 500 m from the tropopause and above. A damping layer was implemented in the uppermost 5 km. The simulation spanned 100 hours, from 00 UTC on 30 September to 04 UTC on 4 October 2015, with outputs recorded at 6-minute intervals.

The initial and boundary conditions of the simulations, including sea surface temperature, were derived from the ERA5 reanalysis. In both D01 and D02, the following model physics schemes were selected: 1) the Kain-Fritsch (KF) convective



scheme (Kain, 2004) for cumulus parameterization; 2) the WRF single moment 6-class (WSM6) scheme (Hong and Lim,
2006) for microphysics; 3) the updated version of the rapid radiative transfer model scheme (RRTMG) (Iacono et al., 2008)
for longwave and shortwave radiation; and 4) the Yonsei University (YSU) planetary boundary layer scheme (Hong et al.,
2006) for the vertical diffusion process. While cumulus parameterizations are theoretically recommended primarily for grid
sizes exceeding 10 km (Skamarock et al., 2008), these schemes also initiate convection at grid sizes smaller than 10 km
(Nasrollahi et al., 2012; Li and Pu, 2009). In the case study of Joaquin, the KF scheme was also applied to the inner domain
to ensure that the simulated hurricane intensity aligned with the IBTrACS dataset.
**4 3-D Stockwell transform (S-transform) wave spectral analysis**
**4.1 The extended S-transform method**
The S-transform, as proposed by Stockwell et al. (1996), is a spectral analysis technique that localizes wave perturbations
in the spatial domain through spectral localization in the frequency domain. The S-transform achieves this by applying a
scalable localizing Gaussian window to the short-time Fourier transform, and it can be applied to any time series or distance
profile to provide localized measurements of wave properties.
For a continuous one-dimensional function of time or distance $h(x)$, the generalized S-transform $S(\tau, f)$ is given as:
$$S(\tau, f) = \int_{-\infty}^{\infty} h(x)\omega_g(x - \tau, f)e^{-i2\pi fx}dx , \qquad (1)$$
where $\tau$ represents the translation in the time (spatial) domain, $f$ is the frequency or wavenumber, and $\omega_g(x - \tau, f)$ is the
normalized Gaussian window:
$$\omega_g(x - \tau, f) = \frac{1}{\sigma\sqrt{2\pi}} e^{\frac{-(x-\tau)^2}{2\sigma^2}} , \qquad (2)$$
In Eq. (2), the normalization factor $\frac{1}{\sigma\sqrt{2\pi}}$ ensures that the integral of the window function equals unity, a prerequisite for
the windowing function employed in the S-transform. The standard deviation, denoted as $\sigma$, is scaled for each frequency using
the formula $\sigma = \frac{c}{|f|}$, where c represents a scaling parameter.
Substituting Eq. (2) into Eq. (1) gives the explicit S-transform:
$$S(\tau, f) = \frac{|f|}{c\sqrt{2\pi}} \int_{-\infty}^{\infty} h(x)e^{\frac{-(x-\tau)^2 f^2}{2c^2}} e^{-i2\pi fx}dx , \qquad (3)$$
The one-dimensional S-transform in Eq. (3) could be extended to three dimensions (Hindley et al., 2019) to localize
wavelengths and directions at every grid point in the 3-D WRF outputs. For function $h(x)$, where $x = (x_1, x_2, x_3)$ is a column
vector describing a 3-D coordinate system, $S(\tau, f)$ can be written as:
$$S(\tau, f) = \frac{1}{(2\pi)^{3/2}} \int_{-\infty}^{\infty} h(x) \left[ \prod_{n=1}^{N} \frac{|f_n|}{c_n} e^{\frac{-(x_n-\tau_n)^2 f_n^2}{2c_n^2}} \right] e^{-i2\pi f^T x}dx , \qquad (4)$$



Here, $\tau = (\tau_1, \tau_2, \tau_3)$ and $f = (f_1, f_2, f_3)$ are column vectors representing spatial translations and spatial frequencies
(inverse of wavelength) in the $x_1, x_2, x_3$ directions, and the $f^T$ denotes the transposed $f$. The scaling factors $c_n = (c_1, c_2, c_3)$
are tuned for each of the three dimensions independently to emphasize different wave properties (Hindley et al., 2016). The S-
transform is used for spectral analysis of gravity waves in diverse geophysical datasets (e.g., Hindley et al., 2016; Wright et
al., 2017; Hindley et al., 2019; Hindley et al., 2020). Here we apply the 3-D S-transform approach of Hindley et al., (2019)
who provide a full description of its implementation in their Sect. 3.

**4.2 Wave properties derived by the 3-D S-transform**


For each WRF output on a 3-D grid $(x, y, z)$, the S-transform in Eq. (4) produces $S(\tau, f) \equiv S(\tau_x, \tau_y, \tau_z, f_x, f_y, f_z)$. We
then collapsed this 6-D object into a 3-D object by only considering the peak of the localized $(f_x, f_y, f_z)$ spectrum for each
location. To do this, for each location in $(\tau_x, \tau_y, \tau_y)$, we find the peak spectral amplitude in the localized spectrum $(f_x, f_y, f_z)$,
which corresponds to the largest amplitude wave at this location. The location of this peak in $(f_x, f_y, f_z)$ tells us the dominant
frequencies, which we record as $F_x(\tau_x, \tau_y, \tau_z)$, $F_y(\tau_x, \tau_y, \tau_z)$, and $F_z(\tau_x, \tau_y, \tau_z)$, respectively.
In our application, the $(\tau_x, \tau_y, \tau_z)$ domain corresponded to the regular grids $(x, y, z)$ of the WRF outputs, i.e.,
$(\tau_x, \tau_y, \tau_z) = (x, y, z)$, so we got the spatial frequencies $f_x \equiv F_x(x, y, z)$, $f_y \equiv F_y(x, y, z)$, $f_z \equiv F_z(x, y, z)$ which are the
inverse of wavelength. Because our WRF grid is cartesian with axes aligned in the zonal, meridional, and vertical directions,
$f_x$, $f_x$, and $f_z$ are simply the zonal, meridional, and vertical wavenumbers $k$, $l$, and $m$, respectively.
Then, the horizontal wavelength is:
$$\lambda_H = \frac{1}{\sqrt{k^2 + l^2}},$$
(5)

and the vertical wavelength is:
$$\lambda_Z = \frac{1}{m},$$
(6)

In Eq. (4), the $c_n$ values of 0.25, 0.5, 0.75, and 1.0 were tested. We empirically selected $c_n = (c_x, c_y, c_z) = (0.5, 0.5, 0.5)$,
which is a compromise between spatial and spectral resolution. In our application, the spatial domain convolution in the S-
transform equation in Eqn. 4 is computed as a frequency-domain multiplication using fast discrete Fourier transform for
efficient computation. As a result, the computed outcomes are discrete.
By simultaneously characterizing the wavenumber $k$, $l$, and $m$, we estimated GWs intrinsic frequencies $\omega$ and phase
speed $c_{ph}$. The GWs intrinsic frequency $\omega$ is calculated from the GW dispersion relation (Fritts and Alexander, 2003):
$$\omega \equiv \sqrt{\frac{N_B^2(k^2 + l^2) + f^2(m^2 + 1/4H^2)}{k^2 + l^2 + m^2 + 1/4H^2}},$$
(7)

where $N_B$ is the Brunt–Väisäla frequency, $H$ is the scale height (~7 km in the stratosphere). $f = 2\Omega \sin(\phi)$ is the Coriolis
frequency, where $\Omega$ is the Earth's rotation rate and $\phi$ is the latitude.
GW intrinsic phase speed $c_{ph}$ is given by:



$$c_{ph} = \frac{\omega}{k^2+l^2+m^2}(k,l,m) \ .$$
(8)

**5 Results**
**5.1 WRF simulations of GWs excited by Hurricane Joaquin (2015)**
The accurate simulation of hurricane intensity is crucial for simulating and analyzing TC-GWs. Figure 1 compares the
simulation results from D01 and D02 with the IBTrACS dataset. Consistent with prior studies (e.g., Jin et al., 2014; Wu et al.,
2018), Hurricane Joaquin's intensity is sensitive to horizontal grid spacing, and the hurricane strengthens as grid spacing
decreases. The maximum surface wind speed (MSFCW) of D01 (12-km grid) underestimated the intensity from 00 UTC, 1
October, and failed to replicate the secondary intensification from 00 UTC, 3 October. The minimum sea level pressure (MSLP)
of D01 also underestimated the intensity but could follow the trend of intensity change.

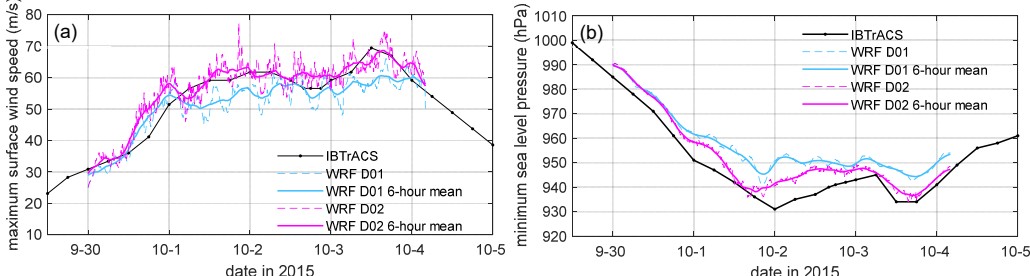

**Figure 1.** (a) Comparison of maximum sustained wind speed (MSW) from IBTrACS with maximum surface wind speed (MSFCW) from
WRF simulation of the outer domain (D01, in blue) and inner domain (D02, in magenta). (b) Comparison of minimum sea level pressure
(MSLP) from IBTrACS with MSLP from WRF simulation of D01 and D02. The simulation outputs every 6 minutes are indicated with thin
dashed lines, and the 6-hour running mean of the outputs every 6 minutes is shown with thick solid lines.
In contrast, the simulated intensity in D02 (4-km grid) exhibits good agreement with the IBTrACS intensity after an initial
spin-up period of approximately 12 hours. D02 successfully replicated the hurricane's rapid intensification until 18 UTC, 1
October, and accurately captured the subsequent weakening, re-intensification, and a second weakening of the hurricane. For
the sake of conciseness, the evaluation of the simulated track and convection strength of the hurricane is presented in the
appendix.
The examples of GWs from D01 and D02, represented by vertical velocities, are compared in Fig. 2, depicting GWs at
30 km altitude during the rapid intensification of Hurricane Joaquin. The GWs simulated with fine and coarse horizontal grid
spacings exhibit notable differences in features and intensities. At 12 UTC on 30 September, GW patterns in D02 are clear,
whereas those in D01 are much less organized, suggesting insufficient spin-up in D01 compared to D02. Afterward, the waves
in both D02 and D01 become clear and organized. Influenced by the easterly flow (not shown), the waves are inhibited
downstream on the western side of the TC, while the wavefronts become more closely packed on the eastern side. In D02,
GWs form tight spirals emanating from the center of the hurricane, with radial horizontal wavelengths of about 30 km at the



depicted time slots in Fig. 2. These spiral waves resemble GWs observed in prior studies (e.g., Chow et al., 2002; Kim and
Chun, 2010; Hima Bindu et al., 2016; Nolan and Zhang, 2017). However, the small-scale waves in D02 are not fully
represented in D01, where radial horizontal wavelengths are approximately 60 km. Unlike D02, D01 did not produce the spiral
GWs that are typically caused by the asymmetry of instabilities rotating along the eyewall (Chow et al., 2002; Nolan, 2020)
or unsteady vortical motions in the inner-core region of TCs (Hendricks et al., 2010). The spiral wave structures in D02 persist
when resampling or averaging the 4-km grid results to the 12-km grid (not shown).

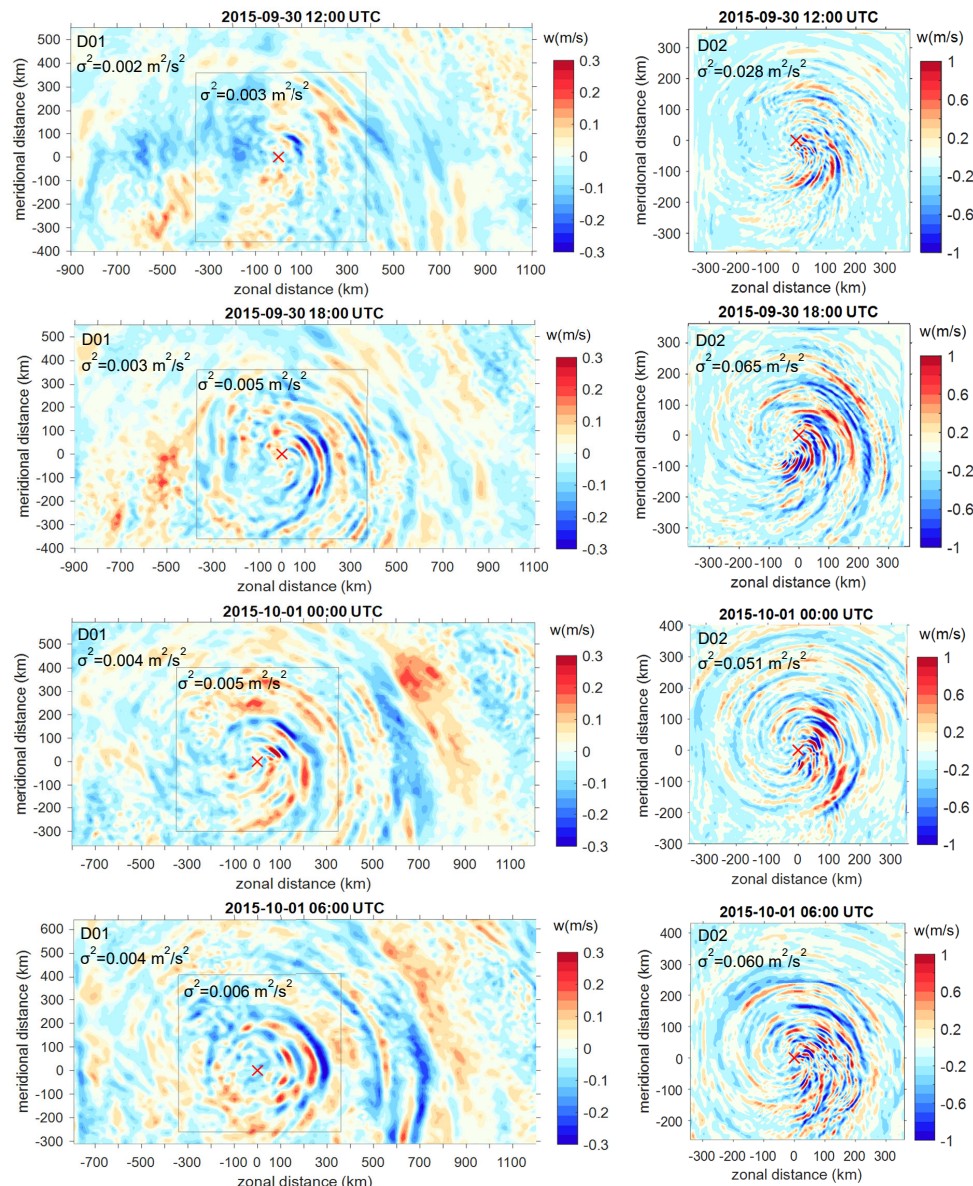






**Figure 2.** Vertical velocities (w, unit: m/s) at 30 km altitude in D01 (left panel) and D02 (right panel). The red crosses indicate the hurricane center. The gray rectangles on the left panel denote the area of D02. The values of the intensity of GWs represented as the variances of vertical velocities ($\sigma^2$) in the domains are labeled. Please notice that different color bar ranges are used for D01 and D02.

The GW intensity (GWI), represented by the variances of vertical velocities ($\sigma^2$), increased as the hurricane intensified. The GWI is much larger in D02 than that in D01. Before D01 fully spun up, the GWI in D02 was ten times as large as that in the same area in D01. Afterward, the GWI in D02 is about four to five times larger than in D01. In the following sections, we focus on analyzing the simulation results from D02 because they are more accurate in terms of hurricane intensity and intensity change and may produce more reliable results of the GWs features associated with the intensity tendency.

**5.2 GW intensity associated with hurricane intensity**

Figure 3 shows the time series of GWI at altitudes of 25, 30, 35, and 40 km. Generally, GWs exhibit higher intensity during the intensification period compared to the weakening period, aligning with previous statistical analyses based on satellite observations (Hoffmann et al., 2018; Wright, 2019). In Fig. 3b, the GWI from 00 UTC, 1 October to 12 UTC, 2 October, is magnified and overlaid with the time series of maximum heating rate ($\partial T/\partial t$) in the upper troposphere (5–15 km) denoted as HR. As depicted in Fig. 3b, considering thermal forcing in convection as one of the primary triggers for GWs, phases of increased HR precede phases of increased GWI, with varying time lag.

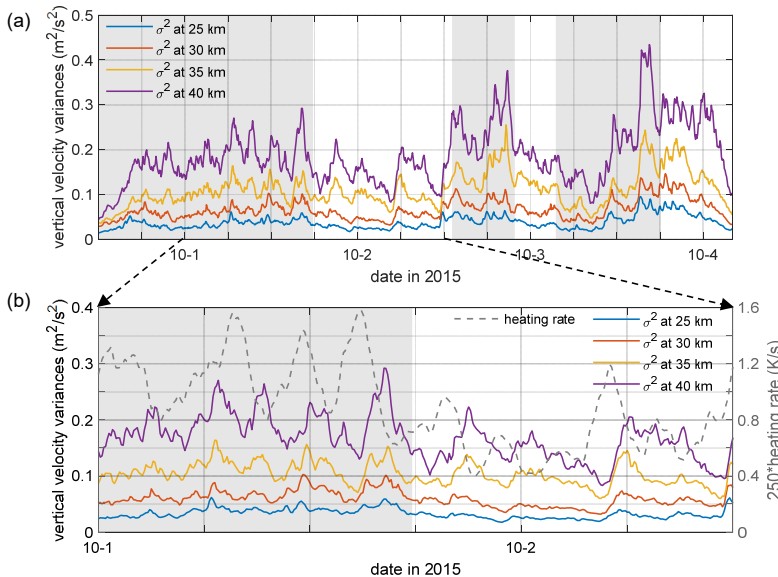

**Figure 3.** (a) Time series of GW intensity (vertical velocity variances $\sigma^2$) on levels of 25, 30, 35, and 40 km from 12 UTC, 30 September to 04:00 UTC, 4 October; (b) Time series of GW intensity overlaid with time series of the maximum heating rate ($\partial T/\partial t$, gray dashed line) at 5–15 km from 00 UTC, 1 October to 12 UTC, 2 October. The intensification periods in (a) and (b) are marked with gray shading.

Using HR as a proxy for thermal forcing, we examine the correlations between thermal forcing and GWI, as well as between thermal forcing and hurricane intensity. Following the approach outlined by Wu et al. (2022), we selected 6-hour



segments at 6-minute intervals, excluding the initial 12 hours of the spin-up period, resulting in 820 6-hour segments. These
segments encompass thermal forcing (represented by HR), hurricane intensity (represented by MSFCW), and stratospheric
GWI (represented by the mean variances of vertical velocities estimated between 20–43 km). Subsequently, Spearman's rank-
order correlation coefficients $\rho$ were calculated between HR and GWI, as well as between HR and MSFCW. To account for
the time shift between the phases of these three variables, we systematically identified the time shift that yielded the largest
time-lagged $\rho_{lag}$ in the 6-hour time series, recording it as the "best" time lag $\tau$.

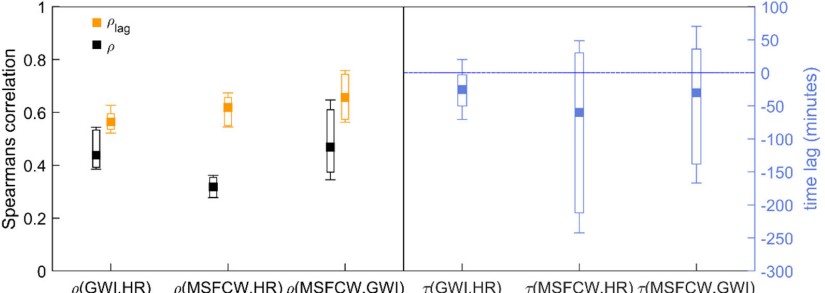


**Figure 4**. Spearman's rank-order correlation coefficients $\rho$ between the GWs intensity (GWI) and the heating rate (HR), the maximum
surface wind (MSFCW) and the HR, and the MSFCW and the GWI are presented on the left. The instantaneous $\rho$ values are depicted in
black, while the time-lagged $\rho_{lag}$ values are shown in orange. All displayed values have undergone a significance test with a confidence level
of 95%. The corresponding time lags ($\tau$) are displayed on the right. The box plot illustrates the minimum, 25th percentile, median, 75th
percentile, and maximum values.
Figure 4 shows the Spearman's correlation coefficients $\rho$ marked in black and the time-lagged $\rho_{lag}$ marked in orange. The
"best" time lag $\tau$ is presented on the right panel of Fig. 4. As depicted in Fig. 4, a moderate correlation is observed among the
three variables, even when not accounting for the time shift. The median values of $\rho$(GWI, HR), $\rho$(MSFCW, HR), and $\rho$(GWI,
MSFCW) are 0.42, 0.33, and 0.50, respectively. Similar to the correlation levels between GWI and HR, and between GWI and
MSFCW at varying altitudes in Wu et al., (2022), all the correlation level increases when considering the time shift between
the three variables. The median values of $\rho_{lag}$(GWI, HR), $\rho_{lag}$(MSFCW, HR), and $\rho_{lag}$(GWI, MSFCW) increase to 0.57, 0.63,
and 0.67, respectively. The time lag $\tau$ between GWI and HR, and MSFCW and HR align with expectations. The negative
median time lag $\tau$(GWI, HR) suggests that GWI follows the variations of HR, which is sensible as GWI depends on the strength
of the thermal forcing.
Compared with Wu et al. (2022), the correlation and time lag between MSFCW and HR, $\rho$(MSFCW, HR), $\rho_{lag}$(MSFCW,
HR), and $\tau$(GWI, HR), are newly incorporated to facilitate comparison with existing research. Time lag $\tau$(MSFCW, HR)
indicates that, in most cases, hurricane intensity changes occur after the HR. Hazelton et al. (2017) noted that strong updrafts
(convective bursts) frequently occur up to 3 hours before TC intensification. These updrafts lead to intense heat release. We
confirmed that heat release is correlated with hurricane intensity change, and the time lag is up to about 240 minutes (4 hours),
consistent with the findings of Hazelton et al. (2017), although the median lag in our Joaquin simulation is about 60 minutes.





The $\tau$(MSFCW, HR) spans both positive and negative values, potentially due to the intricate relationship between hurricane
intensity and heat release.

The correlation between hurricane intensity and GWI is of particular interest in this study. The $\rho_{\text{lag}}$(GWI, MSFCW)

indicates a high correlation between hurricane intensity and GWI. However, the time lag $\tau$(GWI, MSFCW) exhibits a
significant spread, obscuring the sequence of the two activities. Considering that updrafts and thermal forcing are more intense
and stronger during TC intensification (e.g., Guimond et al., 2010, 2016; Hazelton et al., 2017), the GWs triggered during the
intensification period could have distinct properties. Consequently, we calculated the Spearman's correlation coefficients
between GWI and HR, and between GWI and hurricane intensity represented by MSFCW and MSLP separately for the
intensification and weakening periods, as illustrated in Fig. 5a and Fig. 5b. The "best" time lags for the intensification and
weakening periods are presented in Fig. 5c and Fig. 5d.

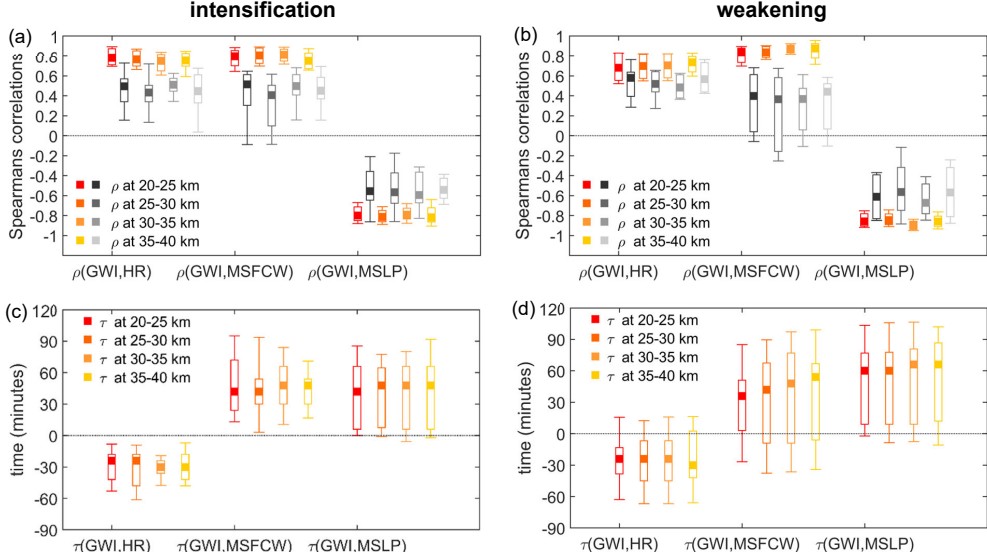


**Figure 5.** Spearman's correlation coefficients and the time lag between variable series. (a) illustrates Spearman's correlation coefficients $\rho$
with instantaneous $\rho$ values marked in black and gray, and time-lagged $\rho_{\text{lag}}$ marked in orange. Only values have undergone a significance
test with a confidence level of 95% are shown. (b) displays the corresponding time lag $\tau$. The box plot illustrates the minimum, 25th
percentile, median, 75th percentile, and maximum values. The left panels correspond to the intensification period, while the right panels
correspond to the weakening period.

Same as for the entire hurricane lifetime in Fig. 4, there are moderate correlations among these variables in the

intensification and weakening scenarios, and the correlation increases to a higher level after considering the time shift. In the
intensification scenario, the spread of time lags is significantly reduced (Fig. 5c), and there is a clear sequence: GWI changes
after HR, and the hurricane intensity changes after GWI. In contrast, in the weakening scenario, the sequence is not clear.
Comparing the time lags with Fig. 4b in Wu et al. (2022), the large spread in the time lag seems induced during the weakening
period. The hypothesis regarding this contrast could be that during the intensification period, the intensive updraft and strong




thermal forcing would generate GWs that propagate quickly to the stratosphere, possibly before the hurricane intensifies. On
the contrary, during the weakening scenario, the vertical propagation of GWs is relatively slow so that the hurricane may
intensify before or after the GWs propagate to the stratosphere, making the time lag between the change of hurricane intensity
and GW intensity unclear.
**5.3 Gravity wave characteristics associated with hurricane intensification**
To explicitly investigate the characteristics of stratospheric GWs during the intensification period, we employed the 3-D
S-transform to estimate the GW spectral properties. As an illustrative example, Fig. 6 presents snapshots of the 3-D GWs,
represented by vertical velocities, and the estimated intrinsic frequency, wavelengths, and intrinsic phase speed at 00 UTC on
1 October 2015. These properties are observed at 35 km altitude and along west-east vertical sections crossing the hurricane
center.
The asymmetric GW patterns, suppressed on the west and compressed on the east side of the hurricane center, are
influenced by the background easterlies. The frequency of the stratospheric GWs excited by Hurricane Joaquin is
approximately one order of magnitude larger than the Coriolis frequency and one order of magnitude smaller than the Brunt-
Väisälä frequency (Fig. 6d–f), indicating a frequency range consistent with mid-frequency GWs. The GWs with relatively
higher frequency correspond to the inner core region, including the eyewall of the hurricane and the region just outside of it
where deep convection occurs actively. These GWs also exhibit shorter horizontal wavelengths (about 20–40 km). The vertical
phase speed exceeds twice the horizontal phase speed, resulting in an upward tilt of the vertical propagation angle (greater
than 45°). The vertical propagation angle in Fig. 6t, overlaid with the horizontal phase velocity represented by the red arrows,
illustrates the outward counterclockwise propagation of waves from the center while simultaneously moving upward, creating
an upward spiral pattern. These waves exhibit faster vertical speeds, enabling them to reach the upper stratosphere without
extensive horizontal propagation from the source. It can be anticipated that wave packets with distinct phase speeds and
complex propagation directions may easily lead to wave superposition. The observed wavy structures, extending from the
inner core to the outer region and from the lower stratosphere to the upper stratosphere, likely reflect the transiency of the
wave sources. This phenomenon is expected and aligns with the characteristics observed in a realistic hurricane case.








**Figure 6**. Properties of the stratospheric GWs excited by Hurricane Joaquin (2015) at 00 UTC on 1 October 2015: (a–c) the simulated vertical velocities w, (d–f) the estimated intrinsic frequency ω, (g–i) horizontal wavelength $\lambda_H$, (j–l) vertical wavelength $\lambda_Z$, (m–o) intrinsic horizontal phase speed $c_{phH}$, (p–r) intrinsic vertical phase speed $c_{phZ}$, and (s–u) vertical propagation angle $\theta_Z$. The vertical propagation angle (t) is overlaid with the horizontal phase velocity (the red arrows). Figures in the left column show the 3D features, figures in the middle column depict the features at 35 km, and those in the right column present a west-east vertical cross-section. Only waves of amplitudes larger than 0.2 m/s and their properties are shown. The red crosses in the middle column indicate the hurricane center. The red crosses in the right column mark the longitude of the hurricane center.

Furthermore, we analyzed the dominant characteristics of stratospheric GWs during the intensification and weakening periods based on the 3-D wave properties from 12 UTC, 30 September, to 04 UTC, 4 October 2015. The analysis is confined to wave properties between 20 and 35 km for two reasons. Firstly, upon comparing the background zonal and meridional winds from the WRF simulations to those from ERA5, we found that the background winds in the stratosphere from the two sources agreed very well below 35 km. However, discrepancies emerged above approximately 35 km due to the absence of some wind reversals in the WRF simulations that were present in the ERA5 winds. These wind reversals could potentially induce wave filtering, implying that wave patterns in the WRF simulations above 35 km might not represent the actual atmosphere. Therefore, waves above 35 km are excluded from the subsequent analysis. Please refer to the appendix for a detailed comparison of simulated background winds and winds from the ERA5 reanalysis. Secondly, we restrict the analysis to estimated wave properties between 20 and 35 km to mitigate potential boundary effects in the wave property analysis.

Figure 7 summarizes the occurrence probability of the intrinsic frequency, horizontal wavelengths, and vertical wavelengths during the whole period of the hurricane from 12 UTC, 30 September to 04 UTC, 4 October 2015, and separately during the intensification and weakening period. The occurrence probability of the intrinsic frequency peaks at $2–3\times10^{-3}$ $s^{-1}$. The hurricane tends to generate GWs with relatively higher intrinsic frequency during the intensification period and relatively lower intrinsic frequency during the weakening period. Specifically, during hurricane intensification, the waves with intrinsic frequency lower than $2\times10^{-3}$ $s^{-1}$ are approximately 14% less compared to the weakening period, whereas the waves with intrinsic frequency between $3–7\times10^{-3}$ $s^{-1}$ show an increase of about 10% compared to the weakening period. The horizontal and vertical wavelengths peak at approximately 40–60 km and 6–8 km, respectively. During the intensification period, the GWs tend to have slightly shorter horizontal wavelengths and longer vertical wavelengths. Figure 8 illustrates this trend from the perspective of wavenumber versus frequency.



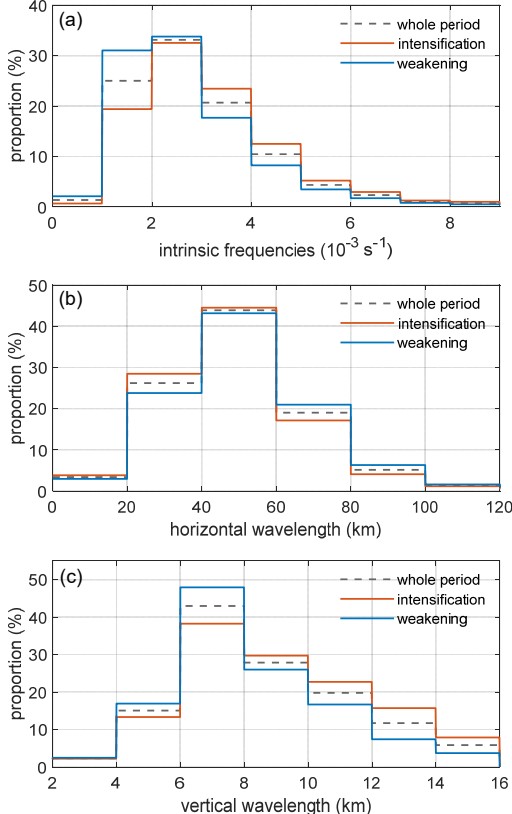

**Figure 7.** The occurrence frequency of stratospheric gravity waves (GWs) properties regarding the change in hurricane intensity. (a) The occurrence probability of intrinsic frequency, (b) horizontal wavelengths, and (c) vertical wavelengths during the entire lifetime and the intensification and weakening periods.

The differences in the occurrence probability of wavenumber versus frequency (intensification minus weakening) indicate that, compared with GWs generated during hurricane weakening, GWs generated during hurricane intensification are characterized by higher frequency ($> 3\times10^{-3}$ s$^{-1}$), shorter horizontal wavelengths (20–40 km), larger horizontal wavenumbers (0.025–0.05 km$^{-1}$), longer vertical wavelengths ($> 8$ km), and smaller vertical wavenumbers (0.06–0.12 km$^{-1}$). As observed from Fig. 9, ascending from the lower stratosphere (20–25 km) to the middle stratosphere (30–35 km), the occurrence possibilities of intrinsic frequency and wavelengths slightly shift to higher frequencies, shorter horizontal wavelengths, and longer vertical wavelengths in both intensification and weakening scenarios. This shift is expected since these waves propagate to higher altitudes faster. As GWs propagate from the lower stratosphere to the middle stratosphere, the contrast between the intensification and weakening scenarios in the above three attributes becomes more prominent.



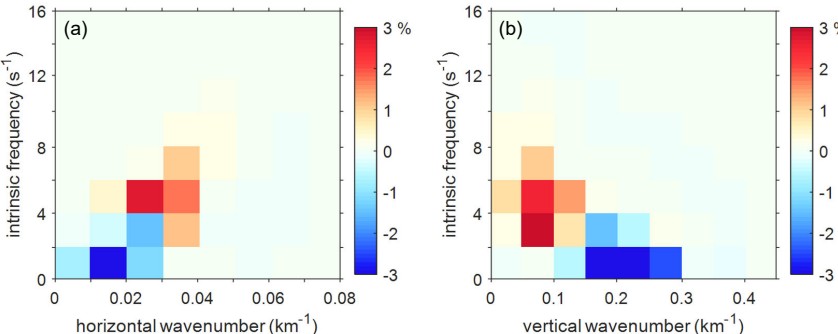

**Figure 8.** Differences in the occurrence probability of wavenumber versus intrinsic frequency during the intensification and weakening period (intensification minus weakening). (a) Horizontal wavenumber versus intrinsic frequency. (b) Vertical wavenumber versus intrinsic frequency.

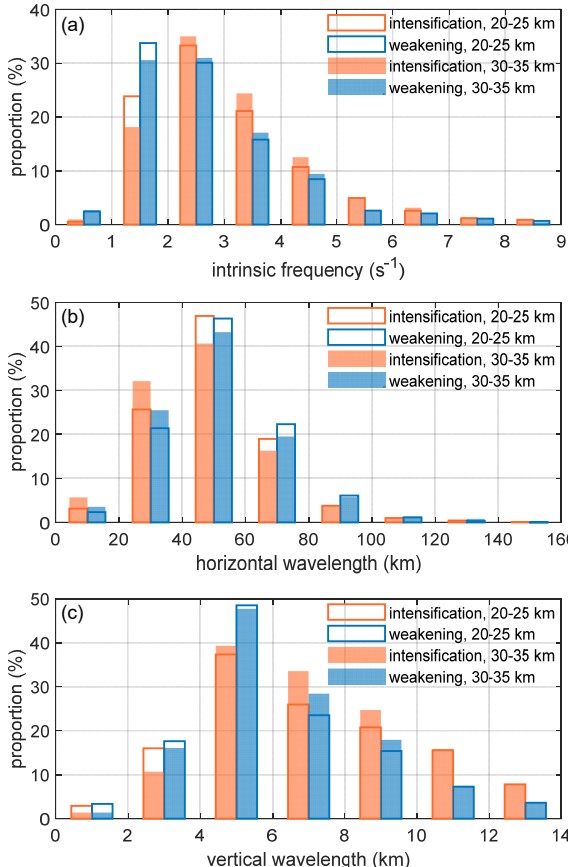

**Figure 9.** Occurrence probability of stratospheric gravity waves (GWs) properties: (a) intrinsic frequency, (b) horizontal wavelengths, and (c) vertical wavelengths separately during hurricane intensification and weakening at the altitude range of 20–25 km and 30–35 km.

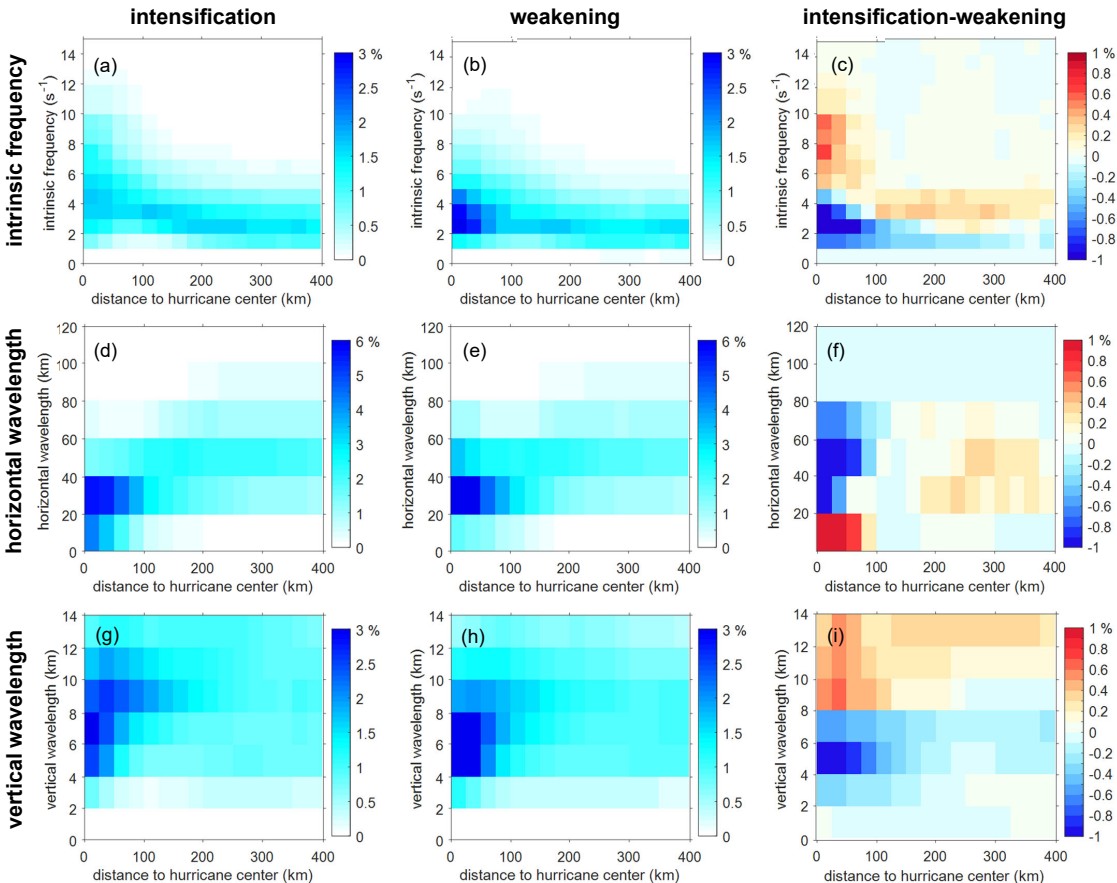

**Figure 10.** The probability density of intrinsic frequency, horizontal wavelengths, and vertical wavelengths during the intensification and weakening period concerning the distance to the hurricane center, and the differences in the occurrence probability density (intensification minus weakening).

Figure 10 illustrates the probability density distribution of intrinsic frequency, horizontal and vertical wavelengths, and their differences concerning the distance to the hurricane center. The waves with relatively higher frequency ($\geq 3\times10^{-3}$ s$^{-1}$), shorter horizontal wavelengths ($\leq 40$ km), and longer vertical wavelengths ($\geq 4$ km) display a higher probability density in the inner core region (approximately 200 km to the hurricane center for the case of Hurricane Joaquin) for both intensification and weakening scenarios. The distinction in the probability density of intrinsic frequency becomes more pronounced in the inner core, particularly for the frequencies exceeding $5\times10^{-3}$ s$^{-1}$. Despite the general tendency for GWs to exhibit shorter horizontal wavelengths during the intensification period, this characteristic diminishes beyond the 100 km radius. GWs generated during the intensification period feature relatively longer vertical wavelengths, a trend that persists within the 400 km radius and is particularly evident in the inner core region as well.

In summary, based on the analyses of the characteristics of the GWs of Hurricane Joaquin, we identified distinctive features in the GWs generated during the intensification of hurricanes. Specifically, the GWs associated with TC intensification



exhibit relatively higher intrinsic frequencies, shorter horizontal wavelengths, and longer vertical wavelengths across the lower
and middle stratosphere. Notably, the differences in wave characteristics are particularly pronounced in the inner core region
of the hurricane, within a radius of approximately 200 km around the center. These GWs that can help separate the hurricane
intensification and weakening periods may indicate the dynamics associated with deep convection in the eyewall and its
immediate surroundings. The identified features of the GWs offer insights that contribute to the interpretation of the findings
presented in Fig. 5. During the intensification period, the time required for the GWs of long vertical wavelengths and high
frequencies to be generated and reach the stratosphere may be shorter than the time needed for the hurricane to intensify in
response to increased convective activities. So, observing the stratospheric GWs with these characteristics could help monitor
the intensification of the hurricane.
**6 Discussion and conclusions**
This study characterized the stratospheric GWs induced by Hurricane Joaquin in 2015 through realistic simulations
using the Weather Research and Forecasting (WRF) model. The simulations encompass wave generation and interactions
with background winds in the actual atmosphere based on ERA5 initial and boundary conditions. Additionally, we employed
the novel 3-D S-transform method to estimate the wave properties. This study emphasizes the specific GW characteristics
and the correlations with hurricane intensification, aiming to provide basic knowledge that could potentially facilitate the
monitoring of hurricane intensification by observing the stratospheric GWs using satellite instruments operating in the
infrared and microwave bands.
Similar as Wu et al. (2022), this study first confirmed the statistically significant correlation between the intensity of the
GWs triggered by TCs and the intensity of TCs observed in long-term satellite data (Hoffmann et al., 2018; Wright, 2019) is
also valid in a specific hurricane case. Additionally, by separating the lifetime of the hurricane into intensification and weaking
period, we found that the pronounced time-lagged correlation between the hurricane intensity and stratospheric GWs primarily
exists during hurricane intensification. This finding suggests the feasibility of monitoring hurricane intensity increases by
observing increased stratospheric GW intensity. Secondly, based on the 3-D wave analyses, we identified distinct
characteristics of the GWs generated during the intensification of Hurricane Joaquin. These features include relatively higher
intrinsic frequency ($> 3 \times 10^{-3}$ $s^{-1}$), shorter horizontal wavelengths ($\leq 40$ km), and longer vertical wavelengths ($\geq 8$ km). It
should be note that the Hurricane Joaquin generated spiral GWs during the period of analysis, which may show relatively
shorter horizontal wavelengths than concentric GWs (Kim and Chun, 2010). The wave packets exhibit rapid vertical speeds,
and the time required for them to propagate up to the stratosphere may be comparable to or even shorter than the time it takes
for the hurricane to intensify in response to increased convection activities. These wave characteristics offer supplementary
information for monitoring and warning of hurricane intensification.



Considering the rapid vertical speeds of these waves, they ascend to the upper stratosphere without extensive horizontal
propagation from the source. A discernible contrast in GW frequencies and wavelengths between hurricane intensification and
weakening scenarios is primarily observed within the inner core ($\leq$ 200 km around the center). Therefore, observing these
distinct GWs, which manifest during hurricane intensification, might necessitate a satellite instrument with extensive coverage
of the inner core region, ideally with high horizontal resolution, such as provided by infrared nadir sounders. Additionally, the
high temporal resolution of the measurements would allow us to better characterize the time evolution of GW and TC intensity
and facilitate the characterization of the GW spectral characteristics. Such high temporal resolution could be achieved by a
satellite instrument operating in a geostationary orbit. Meanwhile, the distinction in GW frequencies and wavelengths between
hurricane intensification and weakening scenarios extends across the lower and middle stratosphere, but it gets even more
evident at higher altitudes. Consequently, a spectral channel focused on higher altitudes may have advantages in identifying
hurricane intensification, provided the GWs can effectively propagate to these elevated altitudes.
Based on the analysis of wave properties outlined above, the 4.3 μm $CO_2$ fundamental band of AIRS on board Aqua, the
Infrared Atmospheric Sounding Interferometer (IASI) instruments on board the European MetOp satellites (Hoffmann et al.,
2013, 2014), or the Cross-track Infrared Sounder (CrIS) instrument on board Suomi-NPP, NOAA-20 and NOAA-21
(Eckermann et al., 2019) hold the potential for observing stratospheric GWs to identify TCs intensification. However,
uncertainties exist in establishing a clear relationship between the observed GWs and the intensity of the TC that triggered the
waves in the real atmosphere. The amplitudes and spectra of the observed waves are significantly influenced by background
winds. Changeable background wind conditions may obscure the distinction between the characteristics of GWs during TC
intensification and weakening. Moreover, the complex thermodynamics of a hurricane are treated as a "black box" in this study.
While the dynamical and thermal processes and changes in TC structure before intensification have been extensively studied
(e.g., Wang and Wang, 2014; Miyamoto and Nolan, 2018), this topic is beyond the scope of our current investigation. The
uncertainties in TC intensity changes resulting from convective activities may introduce additional challenges.
In summary, this study found a high-level time-lagged correlation exists between the stratospheric GWs amplitudes and
the hurricane intensity during the intensification period. Moreover, the stratospheric GWs during hurricane intensification
exhibit relatively higher frequencies, shorter horizontal wavelengths, and longer vertical wavelengths in the inner core region.
The findings of this study further support the feasibility of estimating the intensification of TCs by observing stratospheric
GWs. They may provide further knowledge for optimal utilization of current observation techniques and for planning new
instruments tailored for these specific "target waves." This approach can provide valuable insights for estimating TC
intensification in instances where the top of TCs is obscured by clouds. However, future research, particularly concerning the
influence of background wind conditions, is still needed to better specify the applicable scenarios of this approach.



**Appendices**

**Appendix A Evaluation of the simulated track of Hurricane Joaquin (2015)**

The WRF simulation was conducted for 100 hours from 00 UTC, 30 September to 04 UTC, 4 October 2015. The simulated hurricane track is depicted and compared with the IBTrACS dataset in Fig. A1. An accurate track simulation is necessary to ensure a suitable background for the hurricane's development. As illustrated in Fig. A1, the simulated hurricane track agrees well with the IBTrACS data. The simulation successfully captured the gradual southwestward movement before the track reversed and the subsequent relatively faster northeastward movement. The simulated hurricane center moves slightly slower than the IBTrACS hurricane center in the first 12 hours and faster afterward. The hurricane center moves further toward the northwest compared with the IBTrACS hurricane track after 18 UTC, 3 October.

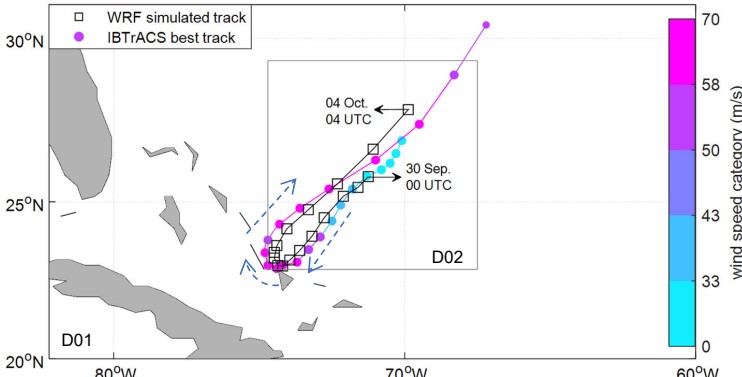

**Figure A1.** The IBTrACS hurricane centers from 00 UTC, 29 September to 12 UTC, 4 October 2015, are shown every 6 hours with colored dots, and the color is assigned according to the IBTrACS maximum sustained wind speed. The WRF simulated hurricane centers during the simulation period (00 UTC, September 30–04 UTC, 4 October 2015) are shown every 6 hours with black squares. The grey rectangle indicates the initial location of the inner domain (D02).

**Appendix B Comparison of WRF simulated background winds with winds from ERA5 reanalysis data**

In order to assess the background winds in the WRF simulation, the mean zonal and meridional winds in the inner domain are compared with those averaged from ERA5 data in the same area. Examples of the comparison at 12 UTC on 1 October 2015 are presented in Figure B1a–b. The background winds from the two sources exhibit good agreement below 30 km, but the discrepancies become more prominent from about 30 km upward. Above 35 km, the meridional wind exhibits a reversal in the ERA5 reanalysis, which is not present in the WRF simulations. Figure B1c–d summarizes the comparison of the winds during the simulation period. Above approximately 35 km, significant differences emerge in the background winds, with frequent instances of opposite wind directions between the simulation and ERA5 reanalysis. The wind reversal observed in the ERA5 reanalysis might induce wave filtering, potentially leading to discrepancies in the wave features between the WRF simulations and the ERA5 reanalysis or the real atmosphere. To ensure the reliability of our analysis, we focused exclusively




on GWs in the lower and middle stratosphere, specifically within the altitude range of 20–35 km, thereby excluding any wave
features questionable in the real atmosphere.

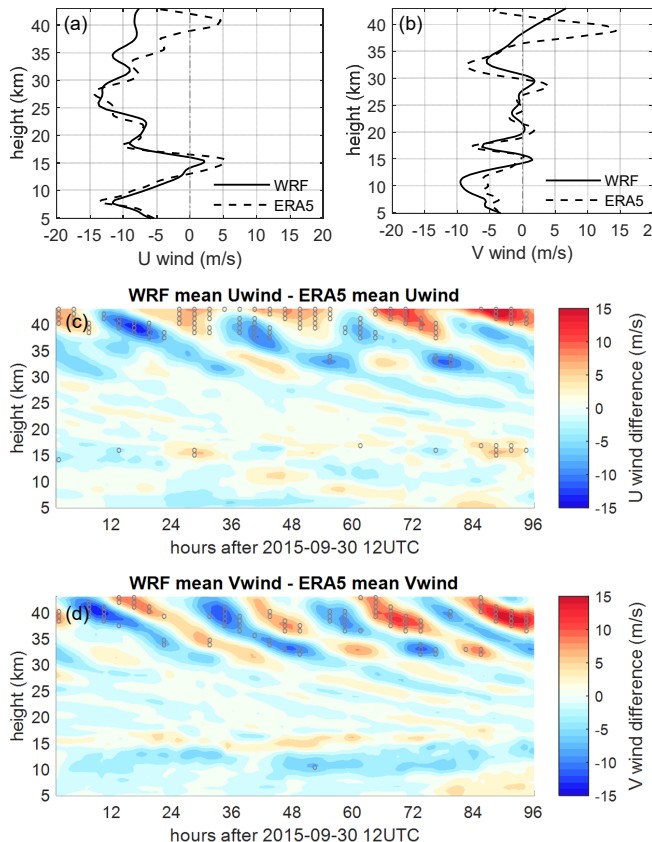


**Figure B1.** Comparison of the mean zonal (U) and meridional (V) winds from the WRF simulation and the ERA5 reanalysis. Examples of
the mean (a) zonal and (b) meridional winds in D02 from WRF simulations and ERA5 reanalysis on 1 October 2015, 12 UTC and time
series of differences of the mean (c) zonal and (d) meridional winds between WRF simulations and ERA5 reanalysis. The gray circles in (c)
and (d) mark opposite wind directions between the WRF simulated and ERA5 winds.
**Data availability.**
The ERA5 reanalysis data (C3S, 2017) were retrieved from the ECMWF Meteorological Archival and Retrieval
System (10.24381/cds.adbb2d47; last accessed: 1 September 2023). IBTrACS data were acquired from the National Centres
for Environmental Information, National Oceanic and Atmospheric Administration (10.25921/82ty-9e16; last accessed: 1
September 2023).



**Author contribution.**

XW and LH drafted the framework of the study. NPH and CJW developed the 3-D Stockwell transform analysis and adapted it to the WRF outputs on 3-D grids. XW, LH, CJW, and BC processed the data, produced all figures, and drafted most of the manuscript. MJA and SK offered key scientific guidance. All authors participated in the editing of the final manuscript.

**Competing interests.**

The authors declare that they have no conflict of interest.

**Acknowledgements.**

XW is supported by the National Natural Science Foundation of China grant no. 41975049, the Basic Strengthening Research Program (Grant 2021-JCJQ-JJ-1058), and the Ground-based Space Environment Comprehensive Monitoring Network (the Chinese Meridian Project II). CJW is supported by the Royal Society University Research Fellowship URF\R\221023 and NERC grants NE/S00985X/1 and NE/V01837X/1. NPH is supported by the NERC Independent Research Fellowship NE/X017842/1. MJA was supported by NASA Weather and Atmospheric Dynamics Program grant no. 80NSSC23K1311. BC is supported by the National Natural Science Foundation of China grant no. 42175046, 42065009, and the Natural Science Foundation of Yunnan Province (Grant 201901BB050045). XW is supported by the Strategic Priority Research Program of the Chinese Academy of Sciences (Grant XDA15021000, XDA15021001). YNW is supported by the second Tibetan Plateau Scientific Expedition and Research Program (Grant 2019QZKK0604).

The computing time and storage were provided by the Juelich Supercomputing Centre.

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
