# Peer review of "Stratospheric gravity waves excited by Hurricane Joaquin in 2015: 3-D characteristics and the correlation with hurricane intensification"

_EGUsphere, 2023_

## Referee Comment (RC2)

Review for "Stratospheric gravity waves excited by Hurricane Joaquin in 2015: 3-D characteristics and the correlation with hurricane intensification,"

By Wu, Hoffman, Wright, Hindley, Alexander, Kalisch, Wang, Chen, Wang, and Lyu.

Summary:

This paper uses a WRF model simulation of Hurricane Joaquin (2015) to assess changes in the properties of gravity waves radiating upward through the stratosphere during the intensification and the steady state phases, and whether observations of these waves could be used to diagnose intensification or weakening.

This paper has several major flaws and many minor ones, and it should be rejected.

Major Comments:

1. I don't see this paper as sufficiently original from Wu et al. (2002). It uses the same very short and poor quality (see below) simulation of Hurricane Joaquin to draw many of the same conclusions. The additional findings about the changes in the properties of the gravity waves between the intensification and steady state changes are new, but some of them are not convincing.

2. The WRF model setup has several strange aspects. First, the minimum resolution of 4 km is on the outer edge of what is believed to be good for simulating rapid intensification. Second, the domain sizes are pretty small (actually not stated in this paper). Third, it uses one-way nesting which makes no sense at all, because 1) in today's computers it is just as easy to run two-way nesting as one-way nesting, so why not do it? And 2) because then the outer boundary condition for d02 is fixed by d01, and gravity waves will not be able to properly radiate out of d02 because of mismatches with d01 (which is like a very similar but still different simulation of the same hurricane). Fourth, the Kain-Fritsch cumulus parameterization is activated on the d02 (nested 4km) grid as well as on d01, which is hard to understand, and defeats part of the purpose of having "cloud-resolving" resolution. The paper states that KF on d02 was used to make the simulation match the best track intensity more closely, which means to me that they had a poor intensity/track simulation without KF on d02, but then discovered that it matched better with it, so they used it. This is also important later because it is not clear if they are accounting for the KF heating tendencies when the compute "HR."

3. The statistical analysis of the time series correlations does not account for degrees of freedom (DOF), and in fact, it artificially inflates the DOF. By making a series of highly overlapping 6-hour time series (each shifted by 6 minutes), you are giving the appearance of many independent data points, but because they are overlapping they are all relying on the same information. In reality there is one time series and even it has less DOF than the number of grid points, which can usually be estimated from the autocorrelation of the series.

4. The author list is suspicious. First, considering the overall effort of the paper, which is statistical and mathematical analyses of the output of a simulation that was previously performed, the author list is strangely long. As required by this journal, the authors provide a section at the end which describes the contributions of the authors. First, it's not clear whether the second "XW" listed is Xue Wu or Xing Wang; I get the impression from the text that it is Xue Wu. Second, two authors of the paper, YW and DL, are not even listed in this section! (Or maybe three depending on Xing Wang.)

Minor comments, by line number.

214-218: GWI should be more carefully defined, especially including what area it is computer over, in both d01 and d02.

222-224: "Maximum heating rate" is not defined and may be not physically meaningful. First, the mathematical expression shown $\partial T/\partial t$, is local (Eulerian) derivative with time, and is only poorly related to moist heating. Second, the "maximum heating rate" sounds like the maximum at any one point, which should not be expected to be physically meaningful because point values can vary wildly in magnitude and location from moment to moment. Something like a volume average over the core of the storm would be meaningful.

263: "more intense and stronger" – redundant

258-260: The fact that changes in MSFCW can precede changes in heating rate is suspicious, and may be caused by the statistical issues noted above.

358-365: It's not clear how the bulk values are computed. Are they storm relative, following the center? Is it just entirely in d02?

375: Here, and repeatedly before, the authors correlate convective "activity" with the properties of the GWs (narrower and faster for intensifying phase). But I am sure they know that what controls their vertical propagation is their wavelengths, and that comes from the horizontal and time scales and the shape of the heating, not from its "intensity."

416: "…treated as a 'black box' in this study." But you have the box! You have the model output and the heating. If you could relate the changes in the structure of the convection, i.e., the individual updrafts as seen in the simulation, to the changes in GW properties, that could alleviate Major Comment #1.

Overall, I would like to say that the authors claims in this paper, that GW properties above TCs change over life cycle and intensification rate, may very well be true. Along with addressing all of the issues above, better simulations and more cases are needed to make a convincing case for publication.

---

## Referee Comment (RC3)

Review of "Stratospheric gravity waves excited by Hurricane Joaquin in 2015: 3-D characteristics and the correlation with hurricane intensification" by Wu et al.

**General comments:**

This work studies the correlations between the stratospheric gravity waves (GWs) excited by TCs and TC intensity for the case of Hurricane Joaquin (2015) using high-resolution WRF simulations and 3D Stockwell transform method. Large time-lagged correlation is found between the hurrican intensification and stratospheric GWs, with different chacteristics between the GWs in the intensifying and weakening stages of the hurrican. Overll, the paper is well organized and well written. The findings are interesting which is thought to be beneficial for the detection of hurrican intensification using satellite observation of stratospheric GWs.

**Major comments:**

1. L157: For the peak spectral amplitude of ($f_x$, $f_y$, $f_z$), do you mean the maximum of sqrt($f_x$^2+$f_y$^2+$f_z$^2)? The authors seem to find out the dominant wave component. Shouldn't this be done for the power spectrum of wave energy rather than the wave frequency?

2. Fig. 3. Are the variables show on this figure derived in the entire D02 or part of it? Are the results sensitive to the domain size? As shown in the right column of Fig. 2, notable GWs are confined in the inner core of the hurricane. Moreover, the maximum heating rate (HR) is shown herein, why not the domain-average HR which should be more relevant to the GWI (variance of vertical velocity in the domain)? More importantly, the HR seems to be the local change of temperature which includes both adiabatic (e.g., temperature advection) and diabatic (e.g., latent heat release) processes? Which process is more relevant to the generation of GWs, the latter one? If this were the case, should it be better show only the diabatic heating rate in the model?

3. About the GW characterisitics in section 5.3. In L291-304, the authors studied various aspects of the GWs. For examlple, high-frequency, short-scale waves are confined in the inner core region while low-frequency ones propagate outward. But there is a lack of

physical explanation for these phenomena. An in-depth discussion of the GW dispersion relationship (combining the wave hoirozntal/vertical scale, phase velocity and intrinsic frequency) will be helpful. Similarly, while the authors found distcint differences between the wave properties in the intensifying and weakening periods, the underlying mechanisms are unclear. For example, why are the wave intrinsic frequency (horizontal wave scale) higher (shorter) in the intensification than in the weakening period?

4. L421-422. While I generally agree with this statement "… *the stratospheric GWs during hurricane intensification exhibit relatively higher frequencies, shorter horizontal wavelengths, and longer vertical wavelengths* …", it is not clear what frequency can be viewed as high or low and what scale is short/long. The differences between the distribution patterns of GWs in the two periods (Fig. 7) are indeed quite small. This may challenge the use of satellite observation of GWs in detecing the hurrican intensification (recalling the last sentence of abstract).

**Minor comments:**

1. L175: *H* is the scale height of stratosphere or troposphere?
2. L251: sensible -> sensitive
3. L343: expected->as expected
4. L393: note->noted
5. Fig.5. Please be more specific with the intensification period in this figure. Is it the three gray shadings in Fig. 3a or only the one in Fig. 3b? So is the weakening period.
6. The title of section 5.3 reads like the authors only studied the GWs during the intensifying period.
7. It's better to add the mean values of each parameter on Fig. 7 and Fig. 9 for comparision. Moreover, to convince the readers, statistical signfiicance tests are needed for these figures.

---

## Author Comment (AC1)

**1. General reply**

*We genuinely thank the reviewers and the editor who made insightful comments that led to improvements to this paper.*

*Please find our point-by-point replies below (in blue color and italics). A revised manuscript with tracked changes has also been uploaded.*

*The major changes made to the manuscript are summarized as follows:*

*1) We added analyses on the relations between the scale of thermal forcing and the wavelengths of gravity waves in Section 4.*

*2) We added the gravity wave ray-tracing study using the GROGRAT model in Section 4, and the GROGRAT model's description was added in Section 3.*

*3) The description of the WRF model is summarized and the detailed model configuration is referred to a previous study.*

*4) We removed the original Fig.4 and Fig.5. Instead, we calculated the correlation and time-lagged correlations between the entire time series of heating rate, gravity wave intensity, and hurricane intensity to demonstrate their correlation and response time generally.*

*5) The abstract, discussion, and conclusion sections have been adjusted based on the revised results section.*

*The first author, Dr. Xue Wu, apologizes for the delay in responding to the reviewers due to her prolonged maternity leave.*

Point-to-point replies to comments:

**Reviewer #1:**

Stratospheric gravity waves excited by Hurricane Joaquin in 2015: 3-D characteristics and the correlation with hurricane intensification

**General comments:**

In the paper "Stratospheric gravity waves excited by Hurricane Joaquin in 2015: 3-D characteristics and the correlation with hurricane intensification" by Xue Wu et al., properties of gravity waves excited by Hurricane Joaquin are studied, using a WRF model simulation. The study is a continuation of work published in the paper Wu et al. 2022, mainly discussing the previous analysis, with focus on division to tropical

cyclone intensification and weakening and on the analysis of wavelengths by 3D Stockwell transform. In general, this topic is important to increase our knowledge about gravity waves and tropical cyclones, with the perspective of improving predictions of rapid intensification of the cyclones.

My major concern about the study is the similarity to the publication Wu et al. 2022. First, texts in the Data and WRF Model Configuration sections are very similar between the papers. This is, of course, understandable, since the data and model configuration are the same. Still, I believe that a few summarizing sentences with a reference to the old paper might be preferable over creating a slightly reformulated duplicate.

*Thank you for the helpful suggestion. In the revised manuscript, we summarized this section and referred the detailed model configuration to Wu et al. (2022).*

Section 5.1 introducing the simulation is mildly extended by showing also the results from the coarse-scale subdomain. However, Fig. 5 presented here as the principal result of Section 5.2 was already published as Fig. S2 of Wu et al. 2022, which is not even mentioned in the submitted paper.

*Compared to Fig. S2 of Wu et al. (2022), the data used to plot Fig. 5 were carefully examined to remove a few time lags and the corresponding correlations that passed the significance test but were suspected as physically unreasonable. The number of values removed was minimal, resulting in Fig. 5 appearing quite similar to Fig. S2 in Wu et al. (2022). Fig. S2 of Wu et al. (2022) was an important figure that illustrated the differences in time lags between stratospheric gravity waves and the hurricane intensity during the intensification and weakening periods. It was added according to constructive suggestions from a reviewer of Wu et al. (2022), but it did not have a chance to be thoroughly discussed. In the revised manuscript, we omitted Fig. 5 and referenced Wu et al. (2022), where discussion with the previous study was necessary.*

And finally, it is not true that this is the first time when 3D Stockwell transform was used for analysis of convective gravity waves, since the method was already used to compute wavelengths in the Wu et al. 2022 paper (although with shorter discussion).

*We remove this statement where it appeared.*

With this in mind, the paper gives the impression of being very short. The main, so-far unpublished, contributions are the plots of the distribution of wavelengths that support the properties already discussed in Wu et al. 2022.

Also, results in the paper are often presented as general properties of tropical

cyclones, although they are based on a single simulation of a single tropical cyclone. More work in this direction (either adding simulations of other cases or a more-detailed discussion of connection to other studies) would increase the validity of the results.

From the positive point of view, the study is comprehensible, with illustrative figures and without extensive number of typos or similar technical problems. Still, I believe the presented work might be considered for publication only after it has been sufficiently extended by more original results.

*In the revised manuscript, we carefully modified the texts, making it explicit that our results are only based on the case of Hurricane Joaquin.*

*Hurricane Joaquin was among the cases that showed a robust correlation between the strong gravity wave activities and hurricane intensification when we did statistical research using ~14 years of satellite observations (Hoffmann et al., 2018; Wu et al., 2022). It is also an appropriate case to study further why the above correlation exists.*

*Your suggestion that we could include more cases is important. When we did the research in Hoffmann et al. (2018), we also found cases in which the above correlation was less significant because the amplitudes of the gravity waves observed by satellites in the stratosphere are also influenced by background winds and the detection capabilities of the satellite instrument. However, including both types of cases might make the goal of this study less focused. So, instead of adding more cases, we added gravity wave ray-tracing analyses to demonstrate the propagation characteristics of the gravity waves separately during the intensification and weakening periods. Further, we relate wavelengths of gravity waves to the scale of thermal forcing, which helps to explain why there are differences in wavelength distribution and wave propagation characteristics during the intensification and weakening period. The discussion section is also extended, and we hope the three studies, Hoffmann et al. (2018), Wu et al. (2022), and this study could tell a complete story.*

*In our future study, we will compare cases of two types, with and without significant correlation between the strong stratospheric gravity waves and hurricane intensification, mainly from the perspective of interactions with background fields and the capability of the Atmospheric Infrared Sounder (AIRS), to estimate under what conditions the correlation could be used as an indicator of hurricane intensification.*

Some specific and technical comments are listed below.

**Specific comments:**

L157: "we find the peak spectral amplitude in the localized spectrum": What does the spectrum look like? Is the Peak amplitude well defined?

[Figure]

**Figure 1.** The specified wave field $h(x, y)$ (background) for which the two-dimensional Stockwell transform (2DST) has been computed. The absolute magnitudes of the localised 2DST wavenumber spectra $|\kappa(k_x, k_y)|$ (foreground) are plotted for three separate locations in **(a)**, **(b)** and **(c)**.

*Hindley et al. (2016) visualize this for the 2-D case. The above figure, Fig.1 of Hindley et al. (2016), shows a specified wave field $h(x, y)$ for which a two-dimensional Stockwell transform has been computed. The absolute magnitude of the localized two-dimensional wavenumber spectrum $|\kappa(k_x, k_y)|$ is plotted for three different example locations. The coefficients of $|\kappa(k_x, k_y)|$ can be directly interpreted as the underlying amplitudes of waves with wavenumbers $k_x$ and $k_y$ at a given location in the specified wave field.*

*The method used in Hindley et al. (2016) and this present study neglects overlapping waves and identifies a single dominant wave for each location in $h(x, y)$. For each such location, we record the complex coefficient of $\kappa(k_x, k_y)$ located at the spectral peak of $|\kappa(k_x, k_y)|$. This yields one complex-valued image $\xi(\tau_x, \tau_y)$, with the same dimensions as the specified wave field $h(x, y)$, which contains the amplitude and phase of the dominant wave at each location. The location of the spectral peak in $|\kappa(k_x, k_y)|$ also gives us the wavenumbers $k_x$ and $k_y$ to which this peak coefficient corresponds, producing two more images $K_x(\tau_x, \tau_y)$ and $K_y(\tau_x, \tau_y)$ which contain the dominant wavenumbers at each location in the specified wave field to which the coefficients of $\xi(\tau_x, \tau_y)$ correspond. Thus, in the three images $\xi(\tau_x, \tau_y)$, $K_x(\tau_x, \tau_y)$, and $K_y(\tau_x, \tau_y)$, we can measure the amplitudes, phases, and wavelengths of the dominant wave features at each location in the specified wave field.*

L163: Are the wavenumbers fx, fy and fz really taken to be wavenumbers k, l and m? In my understanding, this would lead to a different definition of the wavenumbers than e.g. in Fritts and Alexander, 2003 (2 pi factor), so equations 7 and 8 would need

some modifications.

*Thank you for pointing out this error. We estimated the wavenumbers as the inverse of wavelengths. Then, when calculating the intrinsic frequency and the group speed, we converted the wavenumbers by the factor of 2π to suit the definition of the dispersion relation in Fritts and Alexander (2003). This conversion is added to the revised manuscript.*

L220: "Generally, GWs exhibit higher intensity during the intensification period compared to the weakening period". After 10-3, this does not seem to be so clear. Do you have some idea why this could be?

*The wave amplitudes were influenced by background winds. The westerlies got much stronger, particularly above ~30 km, after 3$^{rd}$ October until the end of our simulation at 4:00 UTC on 4$^{th}$ October when the hurricane moved quickly north. Some waves were absorbed, obscuring the possibly larger wave amplitudes during the intensification period. We added a few sentences in the revised manuscript to explain the phenomenon.*

L237: Again, is there always a unique "large" time lag? Some ambiguity in the time lag might align with the large spreads in the plots.

*On some occasions, two or more time lags could result in equally large Spearman's correlation, even after removing time lags that are physically unreasonable, such as those indicating gravity waves appearing hours before thermal forcing. The large spread of time lags can be attributed to the diverse responding time between the variables and the ambiguity in determining the time lags, but the former plays the predominant role.*

**Technical corrections:**

L24 and further: For me, it is very questionable if the 3D Stockwell transform can be considered as "novel", when the original 1D variant was introduced in 1996, with the straight-forward 2D generalisation being developed (for example in the papers of Stockwell) a few years later. Although the first applications on the GWs were using the 1D method only (e.g. around 2010), the use of the 2D transform in a very similar way to the use of the 3D method in the presented study was applied already in Hindley et al. 2016, closely followed by studies using 3D Stockwell transform. Similar time axes can be constructed for many other GW separation methods.

Hindley, N. P., Smith, N. D., Wright, C. J., Rees, D. A. S., & Mitchell, N. J. (2016).

A two-dimensional Stockwell transform for gravity wave analysis of AIRS measurements. Atmospheric Measurement Techniques, 9(6), 2545-2565.

*We remove the statement "novel" where it appeared. We believe this study has innovative results even though it might not be considered the first time that the 3D Stockwell transform method is being intensively used in analyzing the properties of stratospheric gravity waves generated by a hurricane.*

L34: Statement about the past decade is referenced by a 10 years old paper – either add a newer source or reformulate.

*The sentence has been reformulated, and the reference was removed.*

L92: I suggest switching the order of chapters 2 and 3: Chapter 2 currently refers to the simulation that was not mentioned before. (Although merging and significant shortening of the chapters, as discussed above, might be even better.)

*In the revised manuscript, we summarized and shortened the texts describing the WRF model configuration and referred the details to Wu et al. (2022). And we combined the description of data and models into one single section.*

L138, L143, L147: Dots after the equations.

*Fixed.*

L145 and further: The multi-dimensional formulas would be more transparent with a notation for vectors (arrows/bold letters).

*Fixed.*

L148: Missing space after "Here".

*Fixed.*

L164: fx twice.

*Fixed.*

L193: Until 1st October, it seems to me that the D01 domain is even closer to the observation, so I do not see the discussed contrast in this period.

*We rephrased this sentence as: Considering both the MSFCW and the MSLP, the simulated intensity in D02 (4-km grid) exhibits generally better agreement with the IBTrACS intensity.*

L304: "This phenomenon is expected and aligns with the characteristics observed in a realistic hurricane case." Do you have a reference to support this statement?

*This phenomenon refers to the "complex propagation directions may easily lead to wave superposition", not the transient wave source. We deleted the sentence mentioning*

*the transient wave source because it was not directly related to the main results of this study. And we added a reference (Yue et al., 2013) here. Yue et al. (2013) clearly showed gravity wave superposition in their Fig. 2.*

L307 – L308: Math symbols should be in math style.

*Fixed.*

L339 – L340: Possibly put the wavenumbers to a bracket, so that it does not look like new information?

*Fixed.*

L378 – L379 (and elsewhere): Described one simulation only, so use "simulation" instead of "simulations".

*Fixed.*

L387: a specific hurricane case -> the specific hurricane case.

*Fixed.*

**Reviewer #2:**

Review for "Stratospheric gravity waves excited by Hurricane Joaquin in 2015: 3-D characteristics and the correlation with hurricane intensification," By Wu, Hoffman, Wright, Hindley, Alexander, Kalisch, Wang, Chen, Wang, and Lyu.

Summary:

This paper uses a WRF model simulation of Hurricane Joaquin (2015) to assess changes in the properties of gravity waves radiating upward through the stratosphere during the intensification and the steady state phases, and whether observations of these waves could be used to diagnose intensification or weakening. This paper has several major flaws and many minor ones, and it should be rejected.

*This study is part of our ongoing research into the correlations between hurricane intensity and the amplitudes of the gravity waves generated by hurricanes, inspired by Nolan and Zhang (2017). Nolan and Zhang (2017) innovatively proposed a correlation between wave amplitude and hurricane intensity, which initiated several studies exploring the potential of monitoring hurricane intensity by observing the stratospheric gravity waves (Hoffmann et al., 2018; Miller et al., 2018; Tratt et al., 2018).*

*In the first step, we used long-term satellite observations to identify a robust correlation: intensive stratospheric gravity waves tend to appear during hurricane intensification (Hoffmann et al., 2018; Wright, 2019). In the second step, we conducted a realistic model simulation of Hurricane Joaquin 2015 to confirm that this correlation*

*also exists in a single case, indicating that it is not merely a statistical result.*

*This study is the third step, where we aim to explain why the correlation between the strength of stratospheric gravity waves and hurricane intensity is most pronounced during hurricane intensification. This phenomenon is likely related to the distinct characteristics of gravity waves during hurricane intensification, prompting us to analyze properties such as wavelengths. In the revised manuscript, we added additional analyses on 1) the relation of the scale of convection with gravity wavelength and 2) the propagation characteristics of gravity waves to better address the question.*

*Our future research will explore the conditions under which this correlation can indicate hurricane intensification.*

**Major Comments:**

I don't see this paper as sufficiently original from Wu et al. (2002). It uses the same very short and poor quality (see below) simulation of Hurricane Joaquin to draw many of the same conclusions. The additional findings about the changes in the properties of the gravity waves between the intensification and steady state changes are new, but some of them are not convincing.

*No single simulation can meet all research objectives. While our simulation may not be flawless, it effectively serves the goals of this study.*

The WRF model setup has several strange aspects. First, the minimum resolution of 4 km is on the outer edge of what is believed to be good for simulating rapid intensification.

*The optimal resolution for modeling rapid intensification can vary depending on the specific case. In this study, the horizontal grid size of 4 km, along with other model configurations, resulted in an intensity simulation that agrees with IBTrACS data and a deep convection simulation consistent with satellite observations (Wu et al., 2022). Based on our previous experience, we also successfully reproduced the rapid intensification of Hurricane Katrina 2015 using a horizontal resolution of 9 km.*

*The grid spacing significantly impacts the amplitudes and scale of the simulated gravity waves, as well as the hurricane intensity. While the simulated hurricane intensity gets more intensive with decreasing grid spacing (Jin et al., 2014), increasing resolution also yields gravity waves with greater amplitudes (e.g., Wu et al., 2018) and power spectra that skew towards shorter horizontal wavelengths (e.g., Nolan and Onderlinde, 2022; Lane and Knievel, 2005). Increasing resolution is not always*

*beneficial if it leads to unreasonably large hurricane intensity or adds complexity and diversity to convections and wave patterns, making it difficult to interpret the relationship between the hurricane intensity and the waves.*

Second, the domain sizes are pretty small (actually not stated in this paper).

*The inner domain covers an area of 800\*800 km. We apologize for not making this information clear in the manuscript. We intended to add it to the revised manuscript, but considering the suggestion from another reviewer, we decided to summarize the model configuration and refer to our previous study.*

*The location of the inner domain was adjusted as the hurricane moved, and the inner-core region of Hurricane Joaquin was always near or at the center of the inner domain. In Joaquin's case, deep convections occur in the eye wall. Meanwhile, the propagation of convective gravity waves is tiled vertically. Thus, the inner core of the hurricane and the characteristics and propagation of gravity waves could be covered without problems.*

Third, it uses one-way nesting which makes no sense at all, because 1) in today's computers it is just as easy to run two-way nesting as one-way nesting, so why not do it? And 2) because then the outer boundary condition for d02 is fixed by d01, and gravity waves will not be able to properly radiate out of d02 because of mismatches with d01 (which is like a very similar but small different simulation of the same hurricane).

*The choice between one-way and two-way nesting in WRF simulations depends not only on the computational resources available but also on the specific objectives of the simulation. In this study, the detailed analysis focuses on the processes within the inner domain. During the simulation, we were also interested in seeing the differences in the gravity wave characteristics in the inner and outer domains, so we chose not to blend the results from the inner domain back into the outer domain. In preparing this study, we also conducted sensitivity tests that used two-way nesting. In our case, despite the possibility of fine-scale processes interacting with the larger-scale environment, the results within the inner domain were very similar between the one-way and two-way nesting scenarios.*

*When analyzing the results, we excluded the model outputs from the boundary grids (6 grids) to eliminate regions affected by wave reflection.*

Fourth, the Kain-Fritsch cumulus parameterization is activated on the d02 (nested 4km) grid as well as on d01, which is hard to understand, and defeats part of the purpose

of having "cloud-resolving" resolution. The paper states that KF on d02 was used to make the simulation match the best track intensity more closely, which means to me that they had a poor intensity/track simulation without KF on d02, but then discovered that it matched better with it, so they used it. This is also important later because it is not clear if they are accounting for the KF heating tendencies when they compute "HR."

*In our study, we decided to use cumulus parameterization for two main reasons, despite the general best practice of avoiding it at fine grid spacings in WRF. First, the 4 km grid spacing is a balance between our computational constraints and the desired simulation performance. At the edge of the gray zone, this spacing is able to partly resolve convection in Hurricane Joaquin but is not fine enough to fully resolve it. Without cumulus parameterization, the track prediction was satisfactory, capturing the southward and then rapid northward movement of the hurricane center. However, the hurricane intensity in the inner domain was generally somewhat weaker than observed, and the second rapid intensification occurred later than expected. Previous studies have shown that if the hurricane intensity is weaker than observed, it is likely that the latent heat release is also weaker, with the peak of vertical heat located lower than in reality. This can result in discrepancies in gravity wave amplitudes and wave spectra compared to actual observations. We want to improve the intensity simulation to match observational data as closely as possible.*

*The second reason is that, in our case, the use of cumulus parameterization helped maintain computational stability even at the 4 km grid spacing, ensuring the simulation ran smoothly from start to finish. Capturing the full lifecycle of Hurricane Joaquin is important, as it allows us to study the gravity waves excited at all stages of the hurricane. We could still distinguish the differences in gravity wave properties during the intensification and weakening periods by incorporating parameterization.*

*In Section 4.2 of the revised manuscript, where the heating rate was first introduced, we clarified that it was influenced by the use of a cumulus parameterization in the inner domain.*

The statistical analysis of the time series correlations does not account for degrees of freedom (DOF), and in fact, it artificially inflates the DOF. By making a series of highly overlapping 6-hour time series (each shaded by 6 minutes), you are giving the appearance of many independent data points, but because they are overlapping they are all relying on the same information. In reality there is one time series and even it has

less DOF than the number of grid points, which can usually be estimated from the autocorrelation of the series.

*This method, adapted from Wu et al. (2022), aims to visit all the six-hour time series of variable A (e.g., heating rate) to identify which one has the largest correlation with the 6-hour time series of variable B (e.g., gravity wave intensity). This approach helps determine the response time between the two variables while avoiding artificial intervention. Although there is a dependency between the temporally adjacent data points in the time series of a single variable, this is not a problem when our goal is to find the correlation between two variables.*

*As suggested by Reviewer #1, we removed Fig.4 and Fig.5 in the revised manuscript and referred to previous studies when comparing them with earlier results. Thus, this question is no longer a concern for this study. Instead, we calculated the correlation and time-lagged correlations between the entire time series of variables (rather than segments) to generally demonstrate their correlation and response time.*

The author list is suspicious. First, considering the overall effort of the paper, which is statistical and mathematical analyses of the output of a simulation that was previously performed, the author list is strangely long. As required by this journal, the authors provide a section at the end which describes the contributions of the authors. First, it's not clear whether the second "XW" listed is Xue Wu or Xing Wang; I get the impression from the text that it is Xue Wu. Second, two authors of the paper, YW and DL, are not even listed in this section! (Or maybe three depending on Xing Wang.)

*We checked the Author contribution section, ensuring every author's contribution is clearly addressed.*

**Minor comments, by line number.**

214-218: GWI should be more carefully defined, especially including what area it is computed over, in both d01 and d02.

*We added one sentence here to explain how the GWI was calculated.*

222-224: "Maximum heating rate" is not defined and may be not physically meaningful. First, the mathematical expression shown $\partial T/\partial t$, is local (Eulerian) derivative with time and is only poorly related to moist heating. Second, the "maximum heating rate" sounds like the maximum at any one point, which should not be expected to be physically meaningful because point values can vary wildly in magnitude and location from moment to moment. Something like a volume average over the core of

the storm would be meaningful.

*We apologize for not clearly explaining how the 'maximum heating rate' was determined. To get the maximum heating rate, we first calculated the heating rate on 3-D grids. Then, we calculated the mean heating rate at altitudes between 5–15 km (the upper troposphere). The "maximum heating rate" is defined as the largest value of the 2-D volume-averaged heating rate, which effectively represents the strength and location of deep convection. In the Joaquin case, the largest value of the 2-D volume average heating rate always appeared in the eye wall. A supplement movie by Wu et al. (2022) showed this definition accurately captures the location of the deep convection.*

*We clarified the definition in the revised manuscript.*

263: "more intense and stronger" – redundant

*Fixed.*

258-260: The fact that changes in MSFCW can precede changes in heating rate is suspicious and may be caused by the statistical issues noted above.

*That could be one of the reasons. However, under particular conditions, some intensification might occur before updrafts and heating become the dominant factors. We found that favorable environmental conditions can lead to stronger hurricanes, particularly when the hurricane has already been quite strong, which is subsequently supported by increased convective activity, updrafts, and heating. We could not definitively rule out this possibility for all positive $\tau$(MSFCW, HR) values in Fig.4, so we did not exclude them.*

*As mentioned above, Fig.4 has been removed from the revised manuscript, so this question is no longer a concern.*

358-365: It's not clear how the bulk values are computed. Are they storm relative, following the center? Is it just entirely in d02?

*They are storm relative, following the center.*

*As an example, the probability density distribution of intrinsic frequencies concerning the distance to the hurricane center in Fig.10a is derived as follows:*

1) *The intrinsic frequencies are grouped into bins according to their value ranges and the distance (radius) ranges to the hurricane center (0–400 km). The size of the bins is 25 km $\times$ $10^{-3}$ $s^{-1}$.*

2) *The number of elements in each bin $C_i$ is divided by the total number of all elements in all bins N and then multiplied by 100% to calculate the probability*

$v_i$ in each bin:

$$v_i = \frac{C_i}{N} \cdot 100\%$$

3) *The probability $v_i$ in the distance range $r_i$ is then divided by the circular or annular area $\pi \cdot (r_{i+1}^2 - r_i^2)$ to generate the probability density concerning the distance to the hurricane center.*

375: Here, and repeatedly before, the authors correlate convective "activity" with the properties of the GWs (narrower and faster for intensifying phase). But I am sure they know that what controls their vertical propagation is their wavelengths, and that comes from the horizontal and the scales and the shape of the heating, not from its "intensity."

*Our previous studies identified a strong correlation between the **intensity** of stratospheric gravity waves (**not** their wavelength properties) and the intensity change of tropical cyclones, particularly during the intensification phase. This study aims to explain this phenomenon.*

*To enable potential future applications for monitoring or issuing warnings about hurricane intensification through the observation of GWs in the stratosphere, it is crucial to detect intense stratospheric gravity waves before the hurricane reaches the peak intensity, i.e., during the intensification, not after. As illustrated in Fig. 12, a hurricane intensifies immediately and can reach peak intensity within up to 3 hours following thermal forcing ($\Delta t_1$) (Hazelton et al., 2017). Thermal forcing quickly triggers GWs ($\Delta t_2$), while these waves require additional time ($\Delta t_3$) to propagate to the middle stratosphere (~30–35 km). If this combined time $\Delta t_2 + \Delta t_3$ exceeds $\Delta t_1$, stratospheric GWs can not be used for real-time monitoring of hurricane intensification. Therefore, the presence of fast-vertically-propagating waves (with a short $\Delta t_3$) is essential to ensure that $\Delta t_2 + \Delta t_3 \lesssim \Delta t_1$.*

*We added the above explanation to the revised manuscript. Furthermore, in the revised manuscript, we use the ray tracing model, Gravity Wave Regional or Global Ray Tracer (GROGRAT; Eckermann and Marks, 1997; Marks and Eckermann, 1995), to demonstrate the timing of gravity wave vertical propagation based on the gravity wave dispersion relation. These results help to more clearly illustrate the time differences in vertical wave propagation during the intensification and weakening phases.*

[Figure]

*Figure 1. Schematic diagram illustrating the response time of hurricane intensification to thermal forcing and the time interval between thermal forcing and the propagation of gravity waves to the stratosphere.*

*Indeed, in the case of Hurricane Joaquin, the wavelengths of gravity waves are primarily influenced by the scale of thermal forcing. We initially considered this relationship to be a well-established theory and, therefore, did not elaborate on it. However, as you and Reviewer #1 suggested, connecting the wavelengths of gravity waves to the scale of thermal forcing based on our simulation makes a more complete story. Consequently, in the revised manuscript, we have included additional figures and text to demonstrate and explain the relationship between convection scales and the wavelengths of gravity waves.*

416: "…treated as a 'black box' in this study." But you have the box! You have the model output and the heating. If you could relate the changes in the structure of the convection, i.e., the individual updrafts as seen in the simulation, to the changes in GW properties, that could alleviate Major Comment #1.

*As in the response to the previous comment, we added the relation between convection scales and waves in the revised manuscript.*

Overall, I would like to say that the authors claims in this paper, that GW properties above TCs change over life cycle and intensification rate, may very well be true. Along with addressing all of the issues above, better simulations and more cases are needed to make a convincing case for publication.

*We want to thank you once again for your helpful suggestions. We hope we have addressed your questions and believe that the revised manuscript offers improved structure and presentation of results, which better supports our argument.*

**Reviewer #3:**

Review of "Stratospheric gravity waves excited by Hurricane Joaquin in 2015: 3-D characteristics and the correlation with hurricane intensification" by Wu et al.

**General comments:**

This work studies the correlations between the stratospheric gravity waves (GWs) excited by TCs and TC intensity for the case of Hurricane Joaquin (2015) using high-resolution WRF simulations and 3D Stockwell transform method. Large time-lagged correlation is found between hurricane intensification and stratospheric GWs, with different characteristics between the GWs in the intensifying and weakening stages of the hurricane. Overall, the paper is well organized and well written. The findings are interesting which is thought to be beneficial for the detection of hurricane intensification using satellite observation of stratospheric GWs.

**Major comments:**

1. L157: For the peak spectral amplitude of (fx, fy, fz), do you mean the maximum of sqrt(fx^2+fy^2+fz^2)? The authors seem to find out the dominant wave component. Shouldn't this be done for the power spectrum of wave energy rather than the wave frequency?

*We are sorry for the ambiguity in the description of deriving the dominant frequencies. We have rewritten the sentences in L157–159 as: "To do this, for each location in time or spatial domain $(\tau_x, \tau_y, \tau_z)$,ccorresponding to the largest amplitude wave at this location. Then, we record the frequency $(f_x, f_y, f_z)$ at this location as the dominant frequencies $F_x(\tau_x, \tau_y, \tau_z)$, $F_y(\tau_x, \tau_y, \tau_z)$, and $F_z(\tau_x, \tau_y, \tau_z)$."*

2. Fig. 3. Are the variables show on this figure derived in the entire D02 or part of it? Are the results sensitive to the domain size? As shown in the right column of Fig. 2, notable GWs are confined in the inner core of the hurricane. Moreover, the maximum heating rate (HR) is shown herein, why not the domain-average HR which should be more relevant to the GWI (variance of vertical velocity in the domain)? More importantly, the HR seems to be the local change of temperature which includes both adiabatic (e.g., temperature advection) and diabatic (e.g., latent heat release) processes? Which process is more relevant to the generation of GWs, the latter one? If this were the case, should it be better show only the diabatic heating rate in the model?

*The heating rate was initially derived from the temperature tendency for simplicity, as latent heating dominates thermal processes in the eyewall. Estimates suggest that latent heat release can be about five to ten times greater than the sensible heat, though this ratio can vary based on sea surface temperature, humidity, and cyclone intensity. Latent heating also dominates the generation of gravity waves. Therefore, in the revised Fig.3b, we replaced the temperature tendency with the latent heating rate for accuracy. This change does not affect our results or conclusions.*

*The convections modeled in our study are localized and highly transient point sources. So, in the revised Fig.3b, we show the 99th percentile of the vertical velocities and the 99th percentile of the mean latent heating between 5–15 km to denote the strength of the convections. The gravity waves triggered by the convections propagate upwards to the stratosphere and outwards, forming the spirals in D02, as seen in Fig.6. So, we used the domain-averaged gravity wave intensity corresponding to the strength of the convection to investigate their connections.*

3. About the GW characteristics in section 5.3. In L291-304, the authors studied various aspects of the GWs. For example, high-frequency, short-scale waves are confined in the inner core region while low-frequency ones propagate outward. But there is a lack of physical explanation for these phenomena. An in-depth discussion of the GW dispersion relationship (combining the wave horizontal/vertical scale, phase velocity and intrinsic frequency) will be helpful.

*These phenomena conform to the gravity wave dispersion relation. For the mid-frequencies gravity waves ($f"\widehat{\omega}"N_B$) in our study, the dispersion relation can be reduced to:*

$$\widehat{\omega} = N_B \frac{k_h}{m}, \qquad (1)$$

*where $\widehat{\omega}$ is the intrinsic frequency, $N_B$ is the Brunt–Väisäla frequency, and $k_h = \sqrt{k^2 + l^2}$ is the horizontal wave number. And $k$, $l$, and $m$ are the zonal, meridional, and vertical wavenumbers.*

*Equation (1) can be rewritten as:*

$$\lambda_z = \lambda_h \frac{\widehat{\omega}}{N_B}, \qquad (2)$$

*or:*

$$\lambda_z = \frac{\widehat{c_h}}{N_B}, \qquad (3)$$

*where $\lambda_z$ is the vertical wavelength, $\lambda_h$ is the horizontal wavelength, and $\widehat{c_h}$ is the*

*intrinsic horizontal phase speed.*

*As Equation (3) shows, assuming the waves propagation upward and outward in a background with similar $N_B$, same vertical wavelength means similar horizontal phase speed. However, the waves with lower frequencies and thus slower vertical phase speed would take longer to propagate upward to a certain altitude, so the waves could be allowed for a longer time to propagate horizontally.*

*In the revised manuscript, we use the ray tracing model, Gravity Wave Regional or Global Ray Tracer (GROGRAT; Eckermann and Marks, 1997; Marks and Eckermann, 1995), to study the gravity wave propagation based on the gravity wave dispersion relation. The results from the ray tracing could demonstrate these phenomena more straightforwardly.*

*GROGRAT is a well-established tool used for research on gravity wave propagation. The ray tracing equations describe the ray path and refraction along it as follows:*

$$\frac{dx_i}{dt} = \frac{\partial \omega}{\partial k_i},$$

$$\frac{dk_i}{dt} = \frac{\partial \omega}{\partial x_i}, \qquad (4)$$

*Via the dispersion relation, solving these equations is based on U, V, and $N_B$ background fields. In our study, the background fields are provided by the WRF simulation outputs and renewed every six minutes. A (7×7×7)-boxcar filter was applied to smooth the outputs before use to remove the impact of localized gradients in the background field. We traced the gravity waves of amplitudes greater than 0.2 m/s at 32 km backward to the source to calculate their propagation time and the horizontal distance the waves traveled. We excluded the waves whose inferred ray path terminated by GROGRAT at altitudes higher than 18 km. These waves either approached a critical level from above or the wave amplitude vanished because of saturation. Meanwhile, we examined the derived gravity wave phase speeds and frequencies along the inferred ray paths to ensure the sources of gravity waves are the convections of the hurricane, not other instabilities in the stratosphere.*

[Figure]

*Figure 2. The probability density of the propagation time from 32 km downward to 18 km (a–b) and the horizontal propagation distance (c–d) during this propagation time for both the intensification and weakening periods. On each figure panel, the propagation time or horizontal propagation distance from two gravity waves with the same initial vertical wavelengths but distinct initial horizontal wavelengths and intrinsic frequencies are superimposed.*

As we have shown in the manuscript, during the intensification period, gravity waves have relatively longer vertical wavelengths and higher intrinsic frequencies, so Fig.2a shows that during the intensification period, gravity waves may take a relatively shorter time (up to 20 minutes) to propagate from 18 km upward to 32 km. Also, these gravity waves have short horizontal wavelengths, so they may not travel far from their source horizontally (up to 40 km), as seen in Fig.2c. On the contrary, for the weakening period, gravity waves may take up to one hour to propagate from 18 km upward to 32 km (Fig.2b), and the waves may travel up to 80 km during that time (Fig.2d).

For example, the propagation time and distance of two pairs of gravity waves with the same initial vertical wavelengths but distinct intrinsic frequencies and horizontal wavelengths are superimposed. The waves with lower intrinsic frequency and longer horizontal wavelengths could propagate further from the source while propagating upward.

*We added the above analyses in the revised manuscript.*

Similarly, while the authors found distinct differences between the wave properties in the intensifying and weakening periods, the underlying mechanisms are unclear. For example, why are the wave intrinsic frequency (horizontal wave scale) higher (shorter) in the intensification than in the weakening period?

*The gravity wave properties are related to the structure of the convections. Previous linear studies have demonstrated that the vertical wavelengths of gravity waves are about twice the depth of the heating (e.g., Pandya and Alexander, 1999). The change in the buoyancy by a factor of about two at the tropopause suggests that the vertical wavelengths in the stratosphere would be one-half that of the waves in the troposphere. Therefore, there is a correlation between deeper convection and the longer vertical wavelengths. The horizontal wavelengths of the gravity waves are believed to be related to the horizontal scale of the convective system that triggers the waves. For instance, if a convective system spans tens of kilometers horizontally, the gravity waves generated will likely have horizontal wavelengths in the same range. The intrinsic frequencies are also determined by the dimensions and characteristics of the convections. The above relationship could be oversimplified than realistic scenarios when significant shear in the troposphere and non-hydrostatic effects are important.*

*In the revised manuscript, we added the relationship between the depth of the heating and the vertical wavelengths and between the width of the heating and the horizontal wavelength based on our simulations and wave property analyses. We found that during the intensification period, the area of the thermal forcing is slightly more confined than the weakening period. The depth of the heating is deeper during the intensification period than during the weakening period. Therefore, the horizontal wavelengths are slightly shorter, and the vertical wavelengths are slightly longer during the intensification period than during the weakening period.*

*It should be noted that the three mechanisms that generate gravity waves (Beres et al., 2002), 1) deep heating, 2) mechanical oscillator, and 3) obstacle effect, could coexist. Although the deep heating mechanism played the dominant role in generating gravity waves in our study, we also found the dominant oscillating frequency of the updrafts, represented by the 99th percentile of vertical velocity between 5–15 km, is higher ($0.0028 \ s^{-1}$) during the intensification period and lower during the weakening*

*period (0.001 s⁻¹), which may also impact on the intrinsic frequencies of gravity waves. Meanwhile, there are asymmetric wave patterns occasionally in the upstream and downstream areas, which may indicate the obstacle effect of the convective tower.*

4. L421-422. While I generally agree with this statement "… the stratospheric GWs during hurricane intensification exhibit relatively higher frequencies, shorter horizontal wavelengths, and longer vertical wavelengths …", it is not clear what frequency can be viewed as high or low and what scale is short/long. The differences between the distribution patterns of GWs in the two periods (Fig. 7) are indeed quite small. This may challenge the use of satellite observation of GWs in detecting the hurricane intensification (recalling the last sentence of the abstract).

*The last sentence in the abstract might be a little misleading. We did not intend to claim that this study can provide a guideline for judging if a hurricane intensifies by observing the wave patterns from space. In our previous studies, we found a robust correlation between the intensity change of tropical cyclones and the intensity (but not the wave patterns) of stratospheric gravity waves. We hope this study can further explain why the above correlation exists so that this explainable correlation would lay the basis for estimating if a hurricane is intensifying by observing stratospheric gravity wave intensity when it is difficult to use other measurement techniques.*

*We are not clear yet whether the wave wavelengths could also be helpful parameters to indicate hurricane intensification. The wavelength differences during the intensification and weakening period are clear but did not seem very large for Joaquin. Also, the wave patterns in the stratosphere are highly influenced by background conditions. In our future study, we could further study under what conditions the wave patterns could show hurricane intensification and which satellite instruments could detect the differences.*

**Minor comments:**

1. L175: H is the scale height of stratosphere or troposphere?

*To increase computational efficiency when dealing with large volumes of data, we simplify our calculations by not distinguishing between the scale heights in the troposphere and stratosphere, using a uniform value of 7 km. In our case, the $\frac{1}{4H^2}$ is one order of magnitude smaller than $k^2$, $l^2$, or $m^2$, so this simplification is acceptable.*

*The texts here are rewritten as H is scale height (~7 km).*

2. L251: sensible -> sensitive

*Fixed.*

3. L343: expected->as expected

*Fixed.*

4. L393: note->noted

*Fixed.*

5. Fig.5. Please be more specific with the intensification period in this figure. Is it the three gray shadings in Fig. 3a or only the one in Fig. 3b? So is the weakening period.

*The intensification period refers to the period marked by shading in Fig.3a. But we have removed Fig.5.*

6. The title of section 5.3 reads like the authors only studied the GWs during the intensifying period.

*We have modified the title to "Distinct gravity wave characteristics associated with hurricane intensification vs. weakening"*

7. It's better to add the mean values of each parameter on Fig. 7 and Fig. 9 for comparison. Moreover, to convince the readers, statistical significance tests are needed for these figures.

*Thank you for the suggestion. We added the values of the entire period, including the intensification and the weakening phases. These values are considered as the mean values. Additionally, we added violin plots in the revised figures to show the statistics, including the mean and median values. Data used to create the violin plots have passed the statistical significance tests.*

**References:**

*Beres, J. H., Alexander, M. J., and Holton, J. R.: Effects of tropospheric wind shear on the spectrum of convectively generated gravity waves, J. Atmos. Sci., 59, 1805–1824, https://doi.org/10.1175/1520-0469(2002)059<1805:Eotwso>2.0.Co;2, 2002.*

*Eckermann, S. D., and Marks, C. J.: GROGRAT: A new model of the global propagation and dissipation of atmospheric gravity waves, Advances in Space Research, 20, 1253-1256, https://doi.org/10.1016/S0273-1177(97)00780-1, 1997.*

*Fritts, D. C., and Alexander, M. J.: Gravity wave dynamics and effects in the middle*

atmosphere, Rev. Geophys., 41, https://doi.org/10.1029/2001RG000106, 2003.

Hindley, N. P., Smith, N. D., Wright, C. J., Rees, D. A. S., and Mitchell, N. J.: A two-dimensional Stockwell transform for gravity wave analysis of AIRS measurements, Atmos. Meas. Tech., 9, 2545-2565, 10.5194/amt-9-2545-2016, 2016.

Hoffmann, L., Wu, X., and Alexander, M. J.: Satellite Observations of Stratospheric Gravity Waves Associated With the Intensification of Tropical Cyclones, Geophys. Res. Lett., 45, https://doi.org/10.1002/2017GL076123, 2018.

Jin, H., Peng, M. S., Jin, Y., and Doyle, J. D.: An Evaluation of the Impact of Horizontal Resolution on Tropical Cyclone Predictions Using COAMPS-TC, Weather Forecast., 29, 252-270, 10.1175/waf-d-13-00054.1, 2014.

Lane, T. P., and Knievel, J. C.: Some Effects of Model Resolution on Simulated Gravity Waves Generated by Deep, Mesoscale Convection, J. Atmos. Sci., 62, 3408-3419, 10.1175/jas3513.1, 2005.

Marks, C. J., and Eckermann, S. D.: A 3-DIMENSIONAL NONHYDROSTATIC RAY-TRACING MODEL FOR GRAVITY-WAVES - FORMULATION AND PRELIMINARY-RESULTS FOR THE MIDDLE ATMOSPHERE, J. Atmos. Sci., 52, 1959-1984, 10.1175/1520-0469(1995)052<1959:Atdnrt>2.0.Co;2, 1995.

Miller, S. D., III, W. C. S., Yue, J., Seaman, C. J., Xu, S., Elvidge, C. D., Hoffmann, L., and Azeem, I.: The Dark Side of Hurricane Matthew: Unique Perspectives from the VIIRS Day/Night Band, Bulletin of the American Meteorological Society, 99, 2561-2574, 10.1175/bams-d-17-0097.1, 2018.

Nolan, D. S., and Onderlinde, M. J.: The Representation of Spiral Gravity Waves in a Mesoscale Model With Increasing Horizontal and Vertical Resolution, Journal of Advances in Modeling Earth Systems, 14, e2022MS002989, https://doi.org/10.1029/2022MS002989, 2022.

Nolan, D. S., and Zhang, J. A.: Spiral gravity waves radiating from tropical cyclones, Geophys. Res. Lett., 44, 3924–3931, https://doi.org/10.1002/2017GL073572, 2017.

Pandya, R. E., and Alexander, M. J.: Linear stratospheric gravity waves above convective thermal forcing, J. Atmos. Sci., 56, 2434-2446, 10.1175/1520-0469(1999)056<2434:Lsgwac>2.0.Co;2, 1999.

Swenson, G. R., and Liu, A. Z.: A model for calculating acoustic gravity wave energy

and momentum flux in the mesosphere from OH airglow, Geophys. Res. Lett., 25, 477-480, https://doi.org/10.1029/98GL00132, 1998.

Tratt, D. M., Hackwell, J. A., Valant-Spaight, B. L., Walterscheid, R. L., Gelinas, L. J., Hecht, J. H., Swenson, C. M., Lampen, C. P., Alexander, M. J., Hoffmann, L., Nolan, D. S., Miller, S. D., Hall, J. L., Atlas, R., Jr., F. D. M., and Partain, P. T.: GHOST: A Satellite Mission Concept for Persistent Monitoring of Stratospheric Gravity Waves Induced by Severe Storms, Bulletin of the American Meteorological Society, 99, 1813-1828, 10.1175/bams-d-17-0064.1, 2018.

Wright, C. J.: Quantifying the global impact of tropical cyclone-associated gravity waves using HIRDLS, MLS, SABER and IBTrACS data, Q. J. R. Meteorol. Soc., 145, 3013–3039, https://doi.org/10.1002/qj.3602, 2019.

Wu, X., Hoffmann, L., Wright, C. J., Hindley, N. P., Kalisch, S., Alexander, M. J., and Wang, Y.: Stratospheric Gravity Waves as a Proxy for Hurricane Intensification: A Case Study of Weather Research and Forecast Simulation for Hurricane Joaquin, Geophys. Res. Lett., 49, e2021GL097010, https://doi.org/10.1029/2021GL097010, 2022.

Wu, J. F., Xue, X. H., Liu, H. L., Dou, X. K., and Chen, T. D.: Assessment of the Simulation of Gravity Waves Generation by a Tropical Cyclone in the High-Resolution WACCM and the WRF, Journal of Advances in Modeling Earth Systems, 10, 2214-2227, doi:10.1029/2018MS001314, 2018.

Yue, J., Hoffmann, L., and Joan Alexander, M.: Simultaneous observations of convective gravity waves from a ground-based airglow imager and the AIRS satellite experiment, Journal of Geophysical Research: Atmospheres, 118, 3178-3191, 10.1002/jgrd.50341, 2013.